# Off-Axis Flexural Properties of Multiaxis 3D Basalt Fiber Preform/Cementitious Concretes: Experimental Study

**DOI:** 10.3390/ma14112713

**Published:** 2021-05-21

**Authors:** Huseyin Ozdemir, Kadir Bilisik

**Affiliations:** 1Vocational School of Technical Sciences, Gaziantep University, 27310 Sehitkamil-Gaziantep, Turkey; hozdemir2804@gmail.com; 2Nano/Micro Fiber Preform Design and Composite Laboratory, Department of Textile Engineering, Faculty of Engineering, Erciyes University, 38039 Talas-Kayseri, Turkey

**Keywords:** basalt fiber, multiaxis preform concrete, off-axis flexure, flexure energy, winding and casting, damage-tolerant

## Abstract

Multiaxis three-dimensional (3D) continuous basalt fiber/cementitious concretes were manufactured. The novelty of the study was that the non-interlace preform structures were multiaxially created by placing all continious filamentary bundles in the in-plane direction of the preform via developed flat winding-molding method to improve the fracture toughness of the concrete composite. Principle and off-axis flexural properties of multiaxis three-dimensional (3D) continuous basalt fiber/cementitious concretes were experimentally studied. It was identified that the principle and off-axis flexural load-bearing, flexural strength and the toughness properties of the multiaxis 3D basalt concrete were extraordinarily affected by the continuous basalt filament bundle orientations and placement in the pristine concrete. The principle and off-axis flexural strength and energy absorption performance of the uniaxial (B-1D-(0°)), biaxial ((B-2D-(0°), B-2D-(90°) and B-2D-(+45°)), and multiaxial (B-4D-(0°), B-4D-(+45°) and B-4D-(−45°)) concrete composites were considerably greater compared to those of pristine concrete. Fractured four directional basalt concretes had regional breakages of the brittle cementitious matrix and broom-like damage features on the filaments, fiber-matrix debonding, intrafilament bundle splitting, and minor filament entanglement. Multiaxis 3D basalt concrete, particularly in the B-4D structure, controlled the crack phenomena and it was recognized as a more damage-tolerant material than the neat concrete.

## 1. Introduction

Two-dimensional (2D) fabrics and three-dimensional (3D) preforms are used with cementitious matrix to make textile-reinforced concrete (TRC), thus, multiple continuous filamentary TOWs, which is an untwisted bundle of continuous filaments, are incorporated in designed architecture considering the special concrete topology. Numerous research studies on TRC, especially 2D planar fabric-reinforced concrete, were conducted by several researchers [1,2,3,4]. Textile-reinforced concrete was made particularly using the high-strength and high-modulus filamentary bundles such as glass, carbon, basalt, para-aramid, polyethylene and polypropylene. Many 2D planar fabrics (small or large grid size) including leno, non-interlaced or circular fabric [5,6], multiaxis warp knit fabrics [6,7] and 3D orthogonal preforms, have been made with the aid of modified contemporary textile technologies in use for fine concrete applications [6,8]. In addition, a 3D warp-knitted preform has been contemplated for employing in concrete structures [9,10]. Furthermore, several techniques, including pre-stressing, filament winding, pultrusion and hand lay-up were utilized for textile-reinforced concretes [11,12,13]. Many applications of textile-reinforced concretes were identified in civil engineering, including industrial floors, bridge decks, prefabricated panels and rehabilitation (connectors, jackets, confinement, wrapping, repairing, seismic retrofit and anchoring) [14,15,16]. Linear density and fineness of fiber/filament TOWs, fiber placement and directions, fiber volume fraction, fiber tensile strength and modulus, preform architecture, matrix properties and interphase properties between matrix and fibers are the critical dependent and independent parameters of the textile-reinforced concrete materials [17,18]. Recently, functionalized nanomaterials in the form of nanospheres, nanotubes, nanoplatelets and nanofibers were added to textile-reinforced concrete to improve some of the properties, including strength, thermal conductivity and electrical conductivity, electromagnetic shielding effectiveness and sound absorption [19,20,21,22,23].

The most commonly used types of fibers in fiber-reinforced concretes are alkali-resistant glass fiber, high-modulus high-strength para-aramid fiber and high-modulus high-strength polyacrylonitrile (PAN) carbon fiber as well as high-strength polyethylene and PVA fibers. On the other hand, basalt fiber has emerged recently due to its cost effectiveness, comparable strength, high temperature resistance, freeze and thaw performances [24,25,26,27,28]. Basalt fiber has comparable mechanical properties considering the E-glass fibers [29]. It is nontoxic, has good resistance to chemical attack, is noncombustible and low-cost, [30] as well as resistant to infrared and ultraviolet (UV) radiation [31,32,33]. Thus, basalt fiber is an emerging material which is used in constructions and civil applications [34].

It has been reported that axial-reinforced helical circular basalt bar concrete enhanced the bending strength and ductility of neat concrete [35,36]. It was claimed that surface treatment with epoxy/sand mixture on the basalt bar subjected to harsh environments notably affected its bond strength and flexural strength due to strong cohesion bonding between cement and basalt bar [37]. The crack pattern in sand-coated or ribbed basalt fiber-reinforced polymer bars included vertically, diagonally and horizontally oriented cracking and was longer, but the steel bars had horizontal and minor diagonal cracks [38,39]. In addition, it was stated that basalt rebar-reinforced concrete beam showed better shear and flexural cracking compared to those of a steel rebar concrete beam [40,41]. Staple basalt fiber added basalt fiber-reinforced polymer bars under flexural load restrained crack propagation due to the bridging effect of the basalt fiber and enhanced the flexural properties of pristine concrete [42,43]. It was obtained that basalt bar-reinforced concrete with short basalt fibers significantly improved the flexural modulus, but short basalt fibers had an insignificant impact on the concrete’s compressive strength [44]. Another study showed that staple basalt fiber added basalt bar-reinforced concrete enhanced the curvature ductility of the bar and flexure capacity, and crack propagation was properly inhibited via staple fiber bridging mechanism and resulted low crack width [45]. Freeze/thaw cycles on basalt bar concrete resulted crushing mode failure, which was attributed to the increasing of the compressive strength of concrete, but its ductility was not considerably affected [46]. It was found that the strain distribution became nonlinear near the maximum flexure load due to fabric–matrix slippages. Basalt fabric-reinforced concrete shows crushing failure in the compression zone after tensile breakages of basalt fiber [47]. Moreover, carbon and basalt fabric/epoxy-reinforced concrete beams show better load-carrying capacity and ductility due to stiff carbon and flexible basalt fibers in the neat concrete beams [48]. It was claimed that the addition of short basalt fiber in concrete improved the flexural strength, splitting tensile strength and crack resistance as well as ductility [49]. One of the research studies reported that the compressive and splitting tensile strength of recycled waste aggregate glass sand with staple basalt fiber concrete are dependent on the ratio of aggregate glass and volume fraction of basalt fibers [50]. Another study demonstrated that the energy absorption of fiber-reinforced concrete at high strain rate via pure shear after bending test increased with the addition of staple glass and basalt fibers [51].

Peled and Bentur stated that the weave pattern in the flexural capacity of concrete enhanced the bonding and strain-hardening properties by using low modulus polyethylene and polypropylene yarns [52,53]. Single-layer carbon fabric-reinforced concrete with short basalt fiber improved the flexural properties of concrete due to the fiber volume ratio and strong bonding between the fiber and cement matrix [54]. It was identified that short fiber addition to TRC diminished shrinkage-induced crack formation and improved its tensile properties [55]. Basalt fabric-reinforced concrete with staple steel fibers improved the first-crack and post-cracking flexural strength and toughness of the pristine concrete [56,57]. It was found that the amount of short glass fibers in the cement matrix, the number of basalt fabric layers and pre-tension on the basalt fabric are critical parameters which affect the flexural properties of concrete [58]. In addition, it was claimed that basalt fabric and discontinuous polypropylene (PP) fiber was combinedly used as a new type of durable reinforcement in structural members to replace the steel mesh [59]. Staple basalt fiber-reinforced magnesium phosphate cement composite improves the flexural strength and ductility and splitting tensile strength of the neat concrete [60]. Additionally, staple basalt fiber-reinforced concretes decrease the concrete brittleness due to post-peak ductile behavior in tension [61,62]. Another study revealed that the uncoated basalt fiber concrete demonstrates poorer resistance to alkaline medium than zirconia-coated basalt fiber-reinforced concrete [63]. On the other hand, the alkali-resistant basalt fiber in concrete composite increases the flexural strength, splitting tensile strength and compressive strength due to strong bonding between the fiber and matrix [64]. The flexural strength, splitting tensile strength, fracture energy and toughness of high-performance concrete are improved by the addition of staple basalt fiber and polypropylene fibers to pristine concrete [65,66,67]. Staple fibers and pre-tension in basalt fiber concrete considerably enhance the ultimate tensile load, first tensile crack stress and toughness of concrete. For instance, the addition of staple glass and steel fibers to basalt fabric-reinforced concrete augments the number of the cracks, while the crack width and crack spacing were found to be reduced [68].

It was indicated that the fine diameter filament TOW in concrete displays greater load-carrying capacity compared to that of large diameter fiber in textile-reinforced concrete. The reason was largely due to the strong interfacial bonding between the cementitious matrix and high-modulus filament bundle with finer diameter. The finer diameter filament TOWs, owing to their larger surface area, are in contact with the matrix materials more precisely than the large-diameter filament TOWs [69,70]. Other research has demonstrated that the multiple cracks and their distribution on the failed textile concretes are increased due to the engineered design of the unit cell size of the prepreg fabric, and the interfacial bonding between fabric and sand was improved because of sand sticking to the fabric unit cell. In addition, the first-crack load and the eventual load-bearing capacity of the textile-reinforced concrete are increased by the precise design of the fabric unit cell dimensions [71]. The flexural properties of the 3D spacer-reinforced cement composite are affected by Z-fiber density, 3D fabric architecture and directional fiber volume fractions [72,73].

The mechanical properties and electromagnetic interference (EMI) shielding effectiveness of nanoconcrete is improved by addition of the multiwall carbon nanotubes (MWCNT), primarily due to the bridging effect, pore filling and dense calcium silicate hydrate (or C-S-H) material [74,75]. It was claimed that the carbon nanofiber (CNFs, 0.3%) addition in the ultrahigh-performance concrete modifies its microstructure and reduces the porosity of the concrete [76]. The micro and macro crack propagation of fiber-reinforced concrete is controlled by fiber, and its fracture toughness properties are improved via various mechanisms, including fiber bridging, fiber-matrix interfacial debonding, pull-out, sliding or stick-slip and friction [77,78]. However, fiber-cement matrix debonding, bridging and pull-out are the dominant phenomena during failure, which affects the energy dissipation mechanism. This probably causes the slow stable crack-width growth. As a result, the fracture toughness properties of the fiber concrete composite are improved [79]. Recently, Bilisik and Ozdemir found that basalt fiber in 3D preform architecture affects the concrete panel strength, and it is considered as a damage tolerance material due to sequential failure mechanisms occur during panel loading [80].

An artificial neural network (ANN) method was used to model the flexural strength of textile-reinforced concrete (TRC) [81]. Another study pointed out that basalt-reinforced concrete was modelled with the micromechanic damage method, which was indicated via SEM images [82]. Crack opening in discontinuous filament TOW concrete was modelled based on crack stability and proliferation parameters. Fiber length, friction and interfacial shear strength were considered to be the influential factors for the strength and toughness properties of the concrete [83]. The flexural behavior of engineered cementitious composite (ECC) and concrete beams reinforced with basalt fiber-reinforced polymer bars were numerically investigated with the software of ATENA/GID solver [84]. Early-stage stress variation of durable eco-friendly basalt fiber bar concrete during flexure was investigated via image technique, and stress drop, which was called avalanches, resulting from complex interactions between staple basalt fiber and cement matrix, was modeled. It was stated that short fiber volume fraction influences the sudden stress drop and concrete fracture mode [85]. On the other hand, the three-dimensional finite element modeling together with nonlinear analysis of a fiber-reinforced concrete slab interacting with subsoil was studied to identify its mechanical properties and failure mechanism [86].

There have been many types of research on discontinuous high-strength and high-modulus fiber-reinforced concretes, including two-dimensional biaxial fabric-reinforced concretes being made. However, very few studies have been performed on the mechanical properties of multiaxis 3D preform concrete reinforced by continuous filament TOWs with the angular architecture of sparse array. The novelty of this research is that the non-interlace preform structures were multiaxially developed by placing all continious filamentary bundles in the in-plane direction of the preform via developed flat winding-molding method to improve the flexural fracture toughness of the concrete composite. Therefore, the aim of this research was to examine the principle and off-axis flexure properties of multiaxis 3D basalt fiber/cement concrete composites experimentally.

## 2. Materials and Methods

### 2.1. Multiaxis 3D Basalt Preforms and Basalt Concretes

Basalt fiber (technobasalt, UKR) was used to design the multiaxis three-dimensional (3D) preforms for concrete composite. The properties of the basalt fibers are presented in Table 1. As exhibited in Table 1, the basalt fiber properties considering tensile strength, tensile modulus and elongation at break were 2900 MPa, 89 GPa and 3.15%, respectively. Furthermore, the basalt fiber specifications, including the TOW linear density, fiber diameter and density were 4800 tex, 16 μm and 2.80 g/cm^3^, respectively.

Multiaxis 3D basalt preform and multiaxis 3D basalt concrete fabrication steps were broadly explained in the reference [80,89]. However, brief information is provided in this section for the reader. The cementitious matrix including cement (Portland CEM I 52.5, Limak, TR), siliceous fine sand (Pomza export, TR), water and superplasticizer (GLENIUM® 51, BASF, DE) was utilized to produce multiaxis 3D continuous basalt fiber concretes. While fabricating the basalt fiber concrete, general-purpose cement (CEM I 52.5 type Portland cement) was used as a cementitious binder. The natural fine aggregate (100–400 μm) to sustain sufficient bulk rigidity and cohesion of the cementitious matrix was fine silicon dioxide (SiO_2_)-based silica sand in accordance with AS 114131. The density and melting point of silica sand were 2.65 g/cm^3^ and 1750 °C, respectively. A polycarboxylate-type superplasticizer complying with the standard (ASTM C 494 type F) was utilized to improve the rheology and workability of the cementitious matrix. In addition, superplasticizer kept the water/cement ratio low and also attributed to the uniform dispersing of the constituents in the cementitious mixture.

Multiaxis three-dimensional (3D) preform was considered to eliminate crimp exchange and crimp extension (or displacement) in the biaxial fabric. This perhaps caused the stress concentration and led to the ineffective stress transfer in the interlacement region in the biaxial fabric, where the strength performance and fracture mechanism were probably affected [90,91,92]. Proposed preform architectures in this study had sparsely arranged four-layer structures, including two of them angularly arranged in the middle to homogenize the load-bearing performance of the ultimate concretes. The fiber TOW in the multiaxis 3D preform and concrete was continuous form, considering continuity principle [80,89]. 

The “winding and molding method” formed the multiaxis 3D preform, in that this method was proposed by Dr. Bilisik’s continued research efforts since 2000 [80,89]. The preform structure was formed by means of winding the fiber sets around each other, as shown in Figure 1a–c. The constant tension was applied during the winding of the yarn sets by using the in-house building apparatus. Wooden laminate was employed to design and construct the winding apparatus. Its dimensions were 40 (length) cm × 40 (width) cm × 3 (thickness) cm. The fiber TOWs were positioned under pre-tension (4 kg) via precisely drilled holes in both the in-plane and out-of-plane directions. The mold surface was made by using the lubricative layer to prevent fiber breakages during winding and easy demolding after casting. The multiaxis 3D preform structure was made via the defined winding procedure. The non-interlace preform structures were created by placing all the yarns in the in-plane direction of the preform. In the flat winding-molding rig, the continuous basalt fiber TOWs were oriented unidirectionally, bidirectionally and multidirectionally. The preform specifications are presented in Table 2. The fiber volume fraction in the preform increased via using four-plied basalt fiber TOW before the winding process was started. The filament TOW spreading was prevented by twisting the four-plied basalt fibers during the concrete fabrication. The density of the preform and the number of twists were 10 ends/15 cm and 10 turn/m, respectively. The grid size of the B-1D [B-1D-(0°), B-1D-(90°), B-1D-(+45°)], B-2D [B-2D-(0°), B-2D-(90°), B-2D-(+45°)], and B-4D [B-4D-(0°), B-4D-(90°), B-4D-(+45°), B-4D-(−45°)] preforms were 15 mm, 15 × 15 mm and 15 × 15 × 15 × 15 mm, respectively. The layer-to-layer space in the thickness direction of the preform was 6 mm. Flat winding was a common and quick technique for creating multiaxis and multilayered preform structures for concrete. For future endeavor, the multiaxis 3D winding can be integrated with three-dimensional (3D) printing to make one-piece concrete for various applications. For this reason, 3D printing and the multiaxis 3D molding method can be merged, for example, integrating the multifunctional and remotely controlled robotic winding hand and cement mixture and casting via three-dimensional printing.

Multiaxis 3D basalt concretes (B-1D, B-2D, and B-4D) were developed. They were basically made up of four-layered uniaxial (0°), biaxial (0°/90°) and four-direction (90°/±45°/0°) concrete composites. Four-layered basalt fiber concretes were denoted as B-1D, B-2D and B-4D, and their layer sequences were [0°]_4_, [90°/0°]_2_ and [90°/±45°/0°]_1_, respectively (Table 2). B-1D, B-2D and B-4D were four-layered preforms, and their assembling order was [0°]_4_, [90°/0°]_2_ and [90°/±45°/0°]_1_, respectively.

The basalt preforms were consolidated using traditional hand lay-up technique in order to make multiaxis 3D basalt concrete. The manufacturing steps of multiaxis 3D basalt concrete are displayed in Figure 2. The mixture of cementitious concrete was prepared using a Hobart type mixer with a capacity of 20 L. In the beginning, solid ingredients containing Portland cement (735.84 kg/m^3^, 34.51 wt.%) and siliceous fine sand (1030.35 kg/m^3^; 48.33 wt.%) were weighted and blended at 100 rpm for a minute, as shown in Figure 2 schematically. After adding water (364.91 kg/m^3^; 17.12 wt.%) and superplasticizer (0.917 kg/m^3^; 0.04 wt.%) into the dry mixture, they were again blended at a speed of 150 rpm for one minute [93]. Then, in order to have a uniform cementitious matrix, the mixture was mixed once more at 300 rpm for two minutes. Depending on the preform architecture, an in-house-developed flat winding-molding rig was employed to wind basalt continuous TOWs (Figure 1). The matrix mixture was then introduced to the wounded basalt preform in the rig via hand lay-up technique as exhibited in Figure 2. During the matrix insertion process, the flat winding rig was manually shaken to ensure the entire matrix penetration, particularly into the mesh regions in the preform, thus providing strong matrix–TOWs bonding. After casting, basalt fiber concrete was left in the winding-molding rig at room conditions (65 ± 2% relative humidity and 22 ± 2 °C) for 24 h in order to obtain strong consolidation. Thereafter, all basalt concrete composites were separated from the winding rig. In the later stage, the basalt fiber concretes were enfolded with polyethylene film (Reverans Plastik Ltd., Yenimahalle/Ankara, Turkey) prior to curing at 95 ± 5% relative humidity and 23 ± 2 °C for 28 days (Figure 2). Finally, the basalt TOWs concrete panels were ready for the flexure test samples.

### 2.2. Flexural Test for Multiaxis 3D Basalt Preform Concrete

The flexural strength of the multiaxis 3D preform concrete was identified by using a simple beam structure with four-point loading. Though ASTM C78/C78M-18 [94] and ASTM C651–15 [95] are generally considered for the flexure test, they were not followed exactly due to macrogeometric differences between the multiaxis 3D continuous basalt TOWs preform concrete and discontinuous fiber concrete. Pictorial and schematic images of the flexural test are illustrated in Figure 3a–f. The measured flexural test sample sizes were 400 mm (length) × 75 mm (width) × 30 mm (thickness). The off-axis flexural test on the B-1D, B-2D and B-4D samples were conducted in the principal direction (0° and 90°) and ± bias directions (+45° and ‒45°). The dimensions of the principle and off-axis flexural samples were drawn on the top surface of the multiaxis 3D fiber concrete panel. They were cut by using a wet concrete cutting machine which has diamond-tipped saw as shown in Figure 2. Therefore, the prepared samples were coded based on test directions, as B-1D-(0°), B-1D-(90°), B-1D-(+45°), B-2D-(0°), B-2D-(90°), B-2D-(+45°), B-4D-(0°), B-4D-(90°), B-4D-(+45°) and B-4D-(−45°). Data generation on the flexural samples was carried out using a Shimadzu AG- × 100 (JP) testing instrument (SHIMADZU, Kyoto, Japan) which was integrated with Trapezium^®^ software (TRAPEZIUM X, SHIMADZU, Kyoto, Japan) with a 100 kN loading cell. The speed of the cross-head motion was 0.9 mm/min. The flexure tests in the principle and ± bias directions were repeated 5 times and 3 times, respectively. The flexural test was conducted at the standard laboratory atmosphere temperature of 23 ± 2°C and relative humidity of 50 ± 10%. The ASTM C642-13 standard is generally considered to identify the density, absorption and void contents in the multiaxis 3D basalt TOWs concrete [96]. The ASTM D3171-99 standard was followed based on the principle of the rule of mixture in order to obtain the fiber volume fractions of the multiaxis 3D basalt TOWs concrete [97]. The ASTM C78/C78M-18 standard flexural test method for the concrete (such as a simple beam with four-point loading) was followed [94]. Thus, the flexural strength (*σ_ps_*) and initial stiffness (*I_s_*) were calculated using Equations (1) and (2), respectively.
(1)$$\sigma_{ps}=P\times L/b\times d^{2}$$
*I_s_ = P_fc_/δ_fc_*(2)
where *b* is sample width (mm); *d* is sample thickness (mm); *L* is support span length (mm); *P* is maximum applied load (N); *σ_ps_* is flexural strength (MPa); *I_s_* is initial stiffness (N/mm); *P_fc_* is first crack load (N) and *δ_fc_* is first crack displacement (mm).

The multiaxis 3D basalt fiber concrete samples were examined by a scanning electron microscope (SEM, LEO 440VR model, Oxford, UK) before and after the off-axis and principle flexure test. The macro-scale images of the flexure samples were defined via a digital camera (manufacturer, city, state, country) (Nikon D3000 10.2MP Digital SLR Camera with 18–55 mm f/3.5–5.6 G AF-S DX VR Nikkor Zoom Lens, Tokyo, Japan). A steel ruler was used to measure the crack lengths in the fractured flexure samples. A digital caliper (Newman, digital LCD, precision: 0.02 mm, resolution: 0.01 mm, China) was employed to find the crack widths in the fractured samples after the flexure test. In addition, a digital ruler was utilized to measure the crack distance on the multiaxis 3D fiber concrete sample under four-point flexure load [98]. Off-axis pull-out test was applied after the flexure test for identifying the macro-level fiber-cement matrix bonding properties. This study will be published separately in the near future. The area under the entire load-deflection diagram was contemplated for determinations of energy absorption and calculated via MATLAB R2016a (The MathWorks, Inc., Natick, MA, USA) by means of numerical integration and standard plotting tools of MATLAB [99].

## 3. Results and Discussion

### 3.1. Density and Fiber Volume Fraction Results

Table 3 presents the results of the density, fiber volume fraction, absorption and void of the multiaxis 3D basalt TOWs concretes (B-1D, B-2D, and B-4D) and the control concrete (CC). SEM images of the multiaxis 3D concrete composites before the flexural test are exhibited in Figure 4a–i. As depicted in Table 3 and Figure 4, the average fiber weight fraction of all multiaxis 3D concrete samples was 3.783%. Additionally, their mean density and void contents were 2.20% and 19.26%, respectively. In the B-1D (Figure 4a,d), B-2D (Figure 4b,e) and B-4D (Figure 4c,f) concretes, the basalt filaments in the concrete were nearly perfectly bonded by cementitious matrix. Thus, nearly fault-free multiaxis 3D basalt TOWs concrete samples were achieved for the off-axis and principle flexure tests.

### 3.2. Flexural Test Results

Flexural data were achieved for the pristine (CC) and basalt concrete (B-1D-(0°), B-1D-(90°), B-1D-(+45°), B-2D-(0°), B-2D-(90°), B-2D-(+45°), B-4D-(0°), B-4D-(90°), B-4D-(+45°) and B-4D-(−45°)). After the flexural test data obtained from the Shimadzu AG-× 100 (Japan) testing machine, they were transferred to a Microsoft (MS) Excel 2013 spreadsheet via Trapezium^®^ software. Therefore, statistical computations on the raw data including flexural first crack load and displacement, flexural maximum load and displacement, flexural first crack strength and flexural maximum strength, as well as flexural stiffness, were carried out and analyzed. They are introduced in Table 4. Figure 5a–c displays the load-displacement curves for the basalt concrete considering the plain concrete. Figure 6a–f shows stress-displacement curves for the B-1D and B-2D concrete structures. In addition, Figure 7a–d exhibits the stress-displacement curves for B-4D for principle and off-axis directions. Figure 8 illustrates the flexural load-displacement graph of the multiaxis 3D basalt concretes (B-1D-(0°), B-1D-(90°), B-1D-(+45°) and B-2D-(0°), B-2D-(90°), B-2D-(+45°)) after the off-axis four-point flexure test. In addition, Figure 9 illustrates the flexural load-displacement curves of the multiaxis 3D basalt concretes (B-4D-(0°), B-4D-(90°), B-4D-(+45°) and B-4D-(−45°)) after the off-axis four-point flexure test.

In Figure 5a–c, the flexural load-displacement graph of the basalt TOW concretes exhibited remarkably better capacity compared to that of the pristine concrete, particularly due to the introduction of high-performance filamentary TOWs in multiaxis directions. It was identified that flexural load-displacement was significantly dependent on the fiber orientation. For instance, the flexural load of the B-1D-(0°) composite was greater than those of B-2D-(0°) and B-4D-(0°), and the B-4D-(0°) composite was also slightly higher compared to B-2D-(0°) due to the ± bias partial fiber weight fraction in the axial direction. Similar results were obtained for the 90° fiber and ±45° fiber directions and their flexural loads were proportional to their directional partial fiber weight fractions. When the load-displacement curves of the principal (0° and 90°) and ±bias (±45°) fiber directions in the 3D multiaxis basalt fiber concretes were analyzed, four apparent phases were discerned: the elastic phase, where the flexural load was linearly proportional and rose directly up to the first crack in the graph (Figure 5a–c); the numerous initial cracks phase, where multiple cracks in the matrix appeared, especially in ± bias concrete samples (Figure 5c); the strain-hardening phase, where the fibers carried the load exponentially up to the maximum load via diverse mechanisms, such as fiber bridging, complex interphase load transferring through pull-out and probably stick-slip, but it was less obvious in the 90° fiber direction (Figure 5b); and the failure stage, where a catastrophic cementitious matrix and probably multiple minor filament breakages in the 3D fiber concrete occurred (Figure 5a–c).

In Figure 6a–f, the flexural stress-displacement graph of the B-1D-(0°) composite was greater than those of B-1D-(90°), B-1D-(+45°) as well as B-2D-(0°), B-2D-(90°) and B-2D-(+45°) due to the 0° fiber orientation and amount of partial fiber weight fraction in the axial fiber (0° fiber) direction. However, the flexural stress-displacement curves of B-1D-(90°), B-1D-(+45°) and B-2D-(+45°) showed similar curves compared to the pristine concrete.

In Figure 7a–d, the flexural stress-displacement graph of B-4D-(0°) exhibited similar behavior compared to B-4D-(±45°) except B-4D-(90°) in which it showed a slight strain-hardening stage and more ductile material behavior. Material properties in the basalt B-4D concrete were approximately homogeneously distributed via multiaxis fiber orientation throughout the concrete structure.

As depicted in the load-displacement graph of B-1D, B-2D and B-4D basalt TOW concretes (Figure 8 and Figure 9), fiber orientation remarkably affected the sample flexural load-displacement curves. When the 0° fiber changed its positions from bottom (B-4D-(0°)) to the top of the concrete structure (B-4D-(90°)), the flexure load-displacement curves were shifted from strain-hardening to the strain-softening (ductile) material stages. This was identified as a critical finding in the developed multiaxis 3D basalt concrete composites.

### 3.3. Flexural Load-Displacement

Figure 10 shows the average maximum flexural load and displacement and flexural first crack load for the multiaxis 3D continuous fiber concretes. As illustrated in Figure 10 and Table 4, the flexural maximum load of the pristine (CC) concrete was 1374.93 N. The off-axis flexural maximum load of the basalt fiber concretes ranged from 1281.43 N to 4646.59 N for B-1D-(0°, 90°, +45°), from 1542.63 N to 2923.85 N for B-2D-(0°, 90°, +45°) and from 1092.53 N to 3352.22 N for B-4D-(0°, 90°, +45° and −45°).

For the flexural maximum load in principle directions (0° and 90°), B-1D-(0°) was 58.92% higher than B-2D-(0°) and 38.61% higher than for B-4D-(0°), and B-4D-(0°) was 14.65% higher than B-2D-(0°). However, B-4D-(90°) was 29.18% and 14.74% lower than B-2D-(90°) and B-1D-(90°), respectively. B-2D-(90°) was 20.38% higher than B-1D-(90°). For the flexural maximum load in the bias directions (+45° and −45°), B-4D-(+45°) was 51.66% and 82.83% greater than B-2D-(+45°) and B-1D-(+45°), respectively. B-2D-(+45°) was 2.32% greater than B-1D-(+45°). Moreover, B-4D-(+45°) was 5.88% higher than B-4D-(−45°). It was found that flexural load carrying performance in various developed basalt fiber concrete composites was proportional to the fiber orientation and amount of fiber in a particular direction. In addition, B-1D-(0°), B-4D-(0° and ±45°) and B-2D-(0°, 90° and +45°) concrete composites were from 12.20% to 337.95% greater in comparison with the pristine concrete, respectively. It was identified that the basalt fiber placement in the control concrete and the orientation of basalt fiber affected the flexural capacity of the concretes.

In the first crack load, as depicted in Figure 10, the first crack loads of all basalt fiber concretes raised 9.79% relative to the control concrete. In addition, the first crack loads of all the basalt concretes were linearly proportional to their maximum load.

As depicted in Figure 10 and Table 4, the flexural maximum displacement of the pristine (CC) concrete was 0.29 mm. The off-axis flexural maximum displacement of carbon fiber concretes ranged from 0.22 mm to 3.48 mm for B-1D, from 1.58 mm to 3.44 mm for B-2D, and from 0.93 mm to 4.15 mm for B-4D. The average flexural maximum displacement of B-4D was 4.89% higher than B-2D and was 203.65% greater than B-1D. Moreover, the flexure maximum displacement of B-2D was 194.16% greater than that of B-1D concrete. The flexural displacement of the B-4D, B-2D and B-1D concretes were 9.62 times, 9.17 times, and 4.72 times greater compared to that of the pristine concrete, respectively. It was discovered that the basalt fiber placement in the control concrete and the orientation of basalt fiber notably affected the displacement behavior of the concrete. Despite the fact that the neat concrete exhibited a stiff and brittle attitude, the introduction of basalt fiber in the control concrete made it a ductile and tough material. On the other hand, the first crack displacement of all basalt concretes raised 26.55% relative to the control concrete. It was identified that the first crack displacements of all basalt concretes were linearly proportional to their flexure displacement.

### 3.4. Flexural Strength

Off-axis flexure strength, first crack strength and initial stiffness of all multiaxis 3D basalt concretes are shown in Figure 11. As exhibited in Figure 11 and Table 4, the flexure maximum strength of the pristine (CC) concrete was 6.19 MPa. The off-axis flexural maximum strength of the basalt fiber concretes ranged from 5.77 to 20.93 MPa for B-1D, from 6.95 to 13.17 MPa for B-2D and from 4.92 to 15.10 MPa for B-4D.

Regarding flexural maximum strength in the principal directions (0° and 90°), B-1D-(0°) was 58.92% higher than B-2D-(0°) and 38.61% higher than B-4D-(0°), and B-2D-(0°) concrete composite was 12.78% higher than B-4D-(0°). However, B-4D-(90°) concrete was 29.21% and 14.73% greater than B-2D-(90°) and B-1D-(90°), respectively. B-2D-(90°) concrete was 20.45% higher than B-1D-(90°). Regarding the flexural maximum strength in the bias direction (+45° and -45°), B-4D-(+45°) was 51.74% greater than B-2D-(+45°) and 82.86% greater than B-1D-(+45°), and B-2D-(+45°) was 20.51% greater than B-1D-(+45°). In addition, B-4D-(+45°) composite was 5.94% greater than B-4D-(−45°). It was realized that the flexural strength performance of various developed basalt fiber concrete composites was proportional to the fiber orientation and amount of fiber in a particular direction, depending upon the structural fiber architecture. In addition, B-1D-(0°), B-2D-(0° and 90° and +45°) and B-4D-(0° and ±45°) concrete composites had results higher 338.13% and 12.28% compared to that of the control concrete, respectively. Therefore, the basalt fiber placement in the control concrete and the orientation of basalt fiber influenced the flexural strength capacity of the concrete. In first crack strength, as demonstrated in Figure 11, the first crack strength of all basalt TOWs concretes elevated 9.82% compared to that of the neat concrete. It was observed that the first crack strength of basalt TOWs concretes was reciprocal to their flexural strength.

As demonstrated in Figure 11 and Table 4, the flexural initial stiffness of the pristine (CC) concrete was 27.33 MPa/mm. The off-axis flexural initial stiffness of the basalt fiber concretes ranged from 23.46 28.65 MPa/mm for B-1D, from 23.49 to 30.42 MPa/mm for B-2D and from 20.10 to 24.50 MPa/mm for B-4D. The average flexural initial stiffness of the B-4D was 16.92% and 15.41% less than those of the B-2D and B-1D concrete composite, respectively. Furthermore, the flexural initial stiffness of the B-2D concrete was narrowly (1.83%) higher than that of the B-1D concrete. The flexural initial stiffness of the B-4D and B-1D concretes were marginally (20.3% and 5.78%) lower than that of the pristine concrete, respectively. However, the B-1D-(+45°) and B-2D-(+45°) were 4.83% and 11.31% higher compared to the control concrete. The initial stiffness properties of the concrete were probably affected by the structural architecture and the continuous basalt fiber orientations in the multiaxis 3D fiber concrete along with complicated stress transfer at interfacial regions and the fiber-matrix bonding especially during micro-crack initiation and propagation. However, more research studies are required.

### 3.5. Off-Axis Flexural Energy

Average flexure energy data for the multiaxis 3D basalt fiber concretes are demonstrated in Table 5. Figure 12 exhibits representative presentation of areas under the load-displacement graph of the basalt fiber TOW concretes for calculation of flexural first crack energy, flexural maximum load energy, flexural strain-hardening energy, flexural strain-softening energy and flexural total energy. Figure 13a–c shows the off-axis flexural energy-displacement graph of basalt fiber TOW concretes. In addition, Figure 14 exhibits the off-axis flexural energy of all multiaxis 3D basalt fiber concretes. As displayed in Figure 12, the energy absorbed in deflecting a flexure sample at a specified amount is the area under a load-displacement graph in the flexure test. The area under the load-displacement curve of the multiaxis 3D fiber TOW concretes was defined by using relations (3–7) and calculated using MATLAB R2016a (The MathWorks, Inc., Natick, MA, USA) via MATLAB’s numerical integration and standard plotting tools [99], where area calculation was especially achieved by trapezoid method in numerical analysis technique via developed interface algorithm using MATLAB codes.
(3)$$Effc=X_{0}X_{1}X_{2}$$
(4)$$Efml=X_{0}X_{1}X_{3}X_{4}$$
(5)$$Efsh=X_{2}X_{1}X_{3}X_{4}$$
(6)$$Efss=X_{4}X_{3}X_{5}X_{6}$$
(7)$$Eft=X_{0}X_{1}X_{3}X_{5}X_{6}$$
where *Effc* is flexural first crack energy corresponding to the load-displacement in area *X*_0_*X*_1_*X*_2_; *Efml* is flexural maximum load energy corresponding to load-displacement in area *X*_0_*X*_1_*X*_3_*X*_4_; *Efsh* is flexural strain-hardening energy corresponding to load-displacement in area *X*_2_*X*_1_*X*_3_*X*_4_; *Efss* is flexural strain-softening energy corresponding to load-displacement in area *X*_4_*X*_3_*X*_5_*X*_6_ and *Eft* is flexural total energy corresponding to load-displacement in area *X*_0_*X*_1_*X*_3_*X*_5_*X*_6_.

As exhibited in Figure 13a–c, the energy displacement curves became more upward in the order from pristine (CC) to basalt fiber concretes (B-4D, B-2D and B-1D), partly due to the basalt fiber features and straining hardening of the fibers in the concretes. B-1D-(0°), B-2D-(0°) and B-4D-(0° and 90° and ±45°) concretes displayed extraordinarily superior energy absorption capacity compared to those of B-1D-(90° and +45°), B-2D-(+45°) and CC concretes, probably due to the fiber orientation in the concrete structure and that they had strain-hardening energy absorption stages. It was observed that B-1D-(90° and +45°), B-2D-(+45°) and B-4D-(90°) had an insignificant strain-hardening stage (Figure 5b,c). The off-axis fiber energy absorption in B-4D was significant compared to, in particular, B-2D-(+45°) and B-1D-(90° and +45°) concretes due to fiber placement in multiaxial directions and homogeneous fiber distribution throughout the in-plane of the concrete structure.

As exhibited in Figure 14 and Table 5, the flexural maximum load energy of the pristine (CC) concrete was 0.16 J. The basalt fiber concretes (B-1D, B-2D and B-4D) extended from 0.14 to 10.96 J. Regarding flexural maximum load energy in the principal directions (0° and 90°), B-1D-(0°) was 80.26% higher than B-2D-(0°) and 82.67% greater than B-4D-(0°), and B-2D-(0°) was 1.33% higher than B-4D-(0°). However, B-2D-(90°) concrete composite was extraordinarily high compared to B-4D-(90°) and B-1D-(90°) concretes, and B-4D-(90°) was 4.36% higher than B-1D-(90°). Regarding flexural maximum load energy in the bias directions (+45° and −45°), B-4D-(+45°) concrete was 2.95 times greater than B-2D-(+45°) and 39.42-fold greater than B-1D-(+45°), and B-2D-(+45°) was 13.37% greater than B-1D-(+45°). Furthermore, the B-4D-(+45°) was 14.98% lower than B-4D-(−45°). It was found that flexural energy absorption performance of the various developed basalt fiber concrete composites was proportional to fiber orientation and amount of fiber, in particular the direction depending upon structural fiber architecture. Additionally, the energy absorption of B-1D-(0°), B-2D-(0°, 90° and +45°) and B-4D-(0° and ±45°) concrete composites were outstandingly greater compared to the pristine concrete. It was identified that the basalt fiber placement in the concrete structure and the orientation of basalt fiber affected the flexural energy absorption capacity of the concrete. In first crack flexural energy, as shown in Figure 14, the average first crack energy of the B-1D, B-2D and B-4D concretes increased 25%, 31.25% and 12.50% compared to that of the control concrete, respectively.

The flexural strain-hardening energy of the basalt fiber concretes (B-1D, B-2D and B-4D) extended from 0.51 to 10.70 J. Regarding flexural strain-hardening energy in the principal directions (0° and 90°), B-1D-(0°) concrete was 184.80% higher than B-2D-(0°) and 185.12% higher than B-4D-(0°), and B-2D-(0°) composite was almost equal to B-4D-(0°). However, B-2D-(90°) concrete was extraordinarily high (7.45 times) compared to B-4D-(90°). B-1D-(90° and +45°) had no strain-hardening energy stage. Regarding flexural strain-hardening energy in the bias directions (+45° and −45°), B-4D-(+45°) was 15.33% lower than B-4D-(−45°). However, B-4D-(+45°) was 312.88% higher than B-2D-(+45°) concrete composite. It was found that flexural strain-hardening energy absorption performance on various developed basalt fiber concrete composites was proportional to fiber orientation and amount of fiber in a particular direction, depending upon the structural fiber architecture. In addition, B-1D-(0°), B-2D-(0°, 90° and +45°) and B-4D-(0° and ±45°) concretes were extraordinarily greater compared to the pristine concrete.

### 3.6. Off-Axis FTIR Analysis after Flexural Test

Fourier transform infrared (FTIR) spectra of the CC (pristine concrete), basalt fiber, B-1D, B-2D and B-4D basalt fiber concrete composites are shown in Figure 15. Additionally, the FTIR spectra of CC (pristine concrete), basalt fiber and B-4D-(-bias cross-section, +bias cross-section, filling cross-section and warp cross-section) basalt fiber concretes are displayed in Figure 16.

In the FTIR spectrum of the pristine concrete (CC), as demonstrated in Figure 15, the main compound detected in the CC was calcium carbonate, which was supported by several strong bands. To the C–O stretching vibration, 1423 cm^−1^ was assigned. Concerning the other parts of the spectrum, the band at 1083 cm^−1^ may be attributed to Si-O asymmetric stretching vibrations [100]. The bands at 868 and 771 cm^−1^ can be attributed to the -C-C-, -C-H- groups [101,102,103]. The peak extending between 3500 and 3000 cm^−1^ was the absorption peak of the Si–OH and –OH groups. Basalt fiber shows characteristic bands assigned as stretching vibration of Si–O–Si at 890 cm^−1^ [104,105]. A group of intensive bands was observed in the region of 1000–1300 cm^-1^, corresponding to characteristic -Ca-O, Mg-O and Fe-O single bond peaks in the basalt fiber structure [63]. In the 1400–1700 cm^−1^ range a group of bands with small intensities was detected. These bands can be attributed to the stretching vibrations of the Me\O\H bonds, where “Me” stands for metals which are constituents of basalt fiber [63].

In the FTIR spectrum of the B-1D (warp cross-section), B-2D (filling cross-section) and B-4D (warp cross-section) basalt/cementitious concrete composites (Figure 15), a common signal on all developed structures was detected. The peak observed between 3500 and 3000 cm^−1^ bands may be attributed to the deformation vibrations of H_2_O, polycarboxylate, -Si-OH and -Ca-OH groups [106]. In the 3000–2850 cm^−1^ band spacing, the deformation vibrations of -CH2-CH- (aliphatic) groups were found. On the other hand, the signal between 2300 and 2200 cm^−1^ corresponded to −C≡N (nitryl) group. The signals between 1600 and 1300 cm^−1^ were attributed to stretching vibrations of the -C=C-, -C=O and -C=N groups. The signal observed between 1100 and 800 cm^−1^ was assigned to the Si-O single-bond stretching vibrations.

In the FTIR spectrum of the B-4D (-bias cross-section), B-4D (+bias cross-section), B-4D (filling cross-section) and B-4D (warp cross-section), as illustrated in Figure 16, almost similar results were obtained compared to B-1D (warp cross-section) and B-2D (filling cross-section). For instance, the deformation vibrations of H_2_O, polycarboxylate, -Si-OH and -Ca-OH groups were detected between 3500 and 3000 cm^−1^ bands. The deformation vibrations of the -CH2-CH- (aliphatic) groups were observed between 3000 and 2850 cm^−1^ band spacing. The signals between 2300 and 2200 cm^−1^ were attributed to the −C≡N (nitryl) group. The signal observed between 1600 and 1300 cm^−1^ was assigned to stretching vibrations of the -C=C-, -C=O and -C=N groups. The Si-O single-bond stretching vibrations were found between 1100 and 800 cm^−1^. It was considered that fiber placement and fiber orientation in the concrete did not affect their FTIR spectrum. From the FTIR analysis, we comparatively identified a strong physical interaction between cementitious components and basalt fiber (filament TOW). These findings were supported by fractured sample SEM images (Figure 22). Hence, their directional flexural properties were homogeneously distributed throughout the concrete structure. As a result, the B-4D structure controlled the number of cracks, crack width and crack distance and enhanced the flexural energy absorption capacity of the concrete structure. However, more study is required, especially cement–fiber interlaminar regions in the multiaxis 3D fiber concretes.

### 3.7. Flexural Failure Results

Flexure fracture results are exhibited in Table 6. Figure 17 illustrates the crack length and crack numbers in the failed multiaxis 3D basalt fiber concretes after the off-axis four-point flexure test. Furthermore, Figure 18 shows crack width and cracks number in failed multiaxis 3D basalt fiber concretes after the off-axis four-point flexure test.

As displayed in Table 6 and Figure 17 and Figure 18, the pristine concrete demonstrated fatal failure under the four-point flexure load. The off-axis flexural crack width of the basalt fiber concretes ranged from 0.278 to 0.604 mm for B-1D, from 0.392 to 0.774 mm for B-2D and from 0.451 to 0.990 mm for B-4D. The crack distances in B-1D-(0°), B-2D-(0°) and B-2D-(90°) were 129.80, 101.30 and 84.30 mm, respectively. However, the crack distance was 109.90 mm for B-4D-(0°), 76.80 mm for B-4D-(90°), and from 46.10 to 53.80 mm for B-4D-(−45°).

Regarding flexural crack width in principle directions (0° and 90°), B-1D-(0°) concrete was 11.23% higher and 38.99% lower than B-2D-(0°) and the B-4D-(0°) composites, respectively. In addition, B-2D-(0°) was 45.15% lower than B-4D-(0°). On the contrary, the crack width in B-2D-(90°) was 36.27% greater than that of B-4D-(90°), which was near catastrophic failure and B-1D-(90°), which exhibited complete catastrophic failure. Regarding flexural crack width in the bias direction (+45° and −45°), B-4D-(+45°) concrete was 183.42% and 258.63% greater than B-2D-(+45°) and B-1D-(+45°) composites, respectively. The crack width of B-2D-(+45°) was 41.01% greater than that of B-1D-(+45°). However, B-4D-(+45°) was 159.42% higher than B-4D-(−45°). The in-plane crack length in all the developed 3D basalt fiber concrete composites was equal (75 mm). Crack spacing in B-4D-(0°, 90° and −45°) was approximately small compared to those of B-2D-(0° and 90°) and B-1D-(0°) due to the number of the cracks. However, the numbers of cracks in B-4D, and to some degree in B-2D, were greater and multiaxially distributed compared to B-1D. It was deduced that multiaxis 3D basalt fiber concretes, in particular the B-4D structure, managed the number of cracks, crack width, and crack distance during four-point flexure loading. Nonetheless, additional research efforts are expected to identify the plane and sectional crack initiation and propagation under off-axis loading.

### 3.8. Flexural Failure Analysis

Various fractured multiaxis 3D basalt fiber concretes, including B-1D-(0°, 90° and +45°), B-2D-(0°, 90° and +45°), and B-4D-(0°, 90°, +45° and −45°) as well as neat concretes on the top, bottom and cross-sections, and their failure modes after the off-axis flexure test, are exhibited in Figure 19. Figure 20a–l shows some of the pull-out of multiaxis 3D basalt fiber concretes in B-1D-(0° and +45°), B-2D-(0° and 90°) and B-4D-(0°, 90° and +45°) after the off-axis flexural test.

As displayed in Figure 19, in the top and bottom surfaces of the control (CC) concrete, in-plane fatal crack failure was identified, and it demonstrated brittle performance. The CC concrete completely failed to initiate under the flexural loading zone. The control (CC) structure exhibited total through-the-thickness cracking on the front and backside. In the in-plane top (compression side) and bottom (tension side) surfaces of the B-1D-(0° and +45°) structure, minor flexure cracks normal to the warp basalt TOW (0°) were identified. In the out-of-plane back and front sides, flexure failure was initiated, and in a later stage, it was deflected to shear failure, probably due to the modulus differences between the basalt fiber and matrix. The cementitious matrix and fiber near the outer filament bundle surface had debonding failure mode. Further, crack initiation had probably begun and it propagated until through-the-thickness failure was concluded to where multiple warp (0° and +45°) basalt TOWs were not severely damaged. Only minor filament breakages were identified during multiple warp and off-axis pull-outs after the four-point flexure test, as shown in Figure 20a–d. The developed off-axis pull-out test for basalt concrete after flexure test for this purpose is planned for a separate paper to be published in the near future. The in-plane top (compression side), bottom (tension side) surfaces and the out-of-plane through-the-thickness of the B-1D-(90°) structure showed catastrophic flexure crack failures due to a parallel loading direction in the warp fiber. The in-plane top (compression side), bottom (tension side) surfaces and the out-of-plane through-the-thickness of the B-2D-(0°, 90°and +45°) structure had minor flexure cracks where multiple warp and filling (0° and 90°) carbon TOWs were not harshly damaged. Only minor filament breakages were observed during multiple off-axis pull-outs after the flexure test, as shown in Figure 20e–h. The in-plane top (compression side), bottom (tension side) surfaces and the out-of-plane through-the-thickness of the B-4D-(0° and ±45°) structure had minor flexure cracks where multiple warp and filling (0°, 90° and ±45°) basalt TOWs were not severely damaged. Only minor filament breakages were observed during the multiple off-axis pull-outs after the flexure test as shown in Figure 20i–l. The in-plane top (compression side), bottom (tension side) surfaces and the out-of-plane through-the-thickness of the B-4D-(90°) structure showed large flexure cracks opening due to fiber placement and orientation in the structure architecture.

The fracture mechanism probably comprised regional matrix fracture, matrix-filament bundle delamination for each basalt TOW set in the concrete, filament TOW bridging because of tensile strengthening by means of principle (orthogonal) fibers, partly complex shearing via particularly off-axis yarns (non-orthogonal) and shortly after, outer filament pull-out (basically tensile) and stick-slip in the outer filament-matrix regions, which led to interfacial shear strength between fiber and matrix or core (central) filament–filament frictions, which led to complicated interfacial frictional strength, including telescopic movement of filament bundles upon filament-to-filament breakages. Thus, the warp, filling and ± bias yarns carried the flexure load via a complex load distribution of the matrix. Nonetheless, additional studies are required to understand the complicated crack initiation-propagation, considering some types of stress concentration or delaying micro-crack in the through-the-thickness for each fiber set during flexure loading.

### 3.9. Failure Analysis by SEM Micrograph

Figure 21a–d depicts the SEM images of internal fracture surface of the B-1D and B-2D concrete materials. The core filaments of the basalt TOWs in the B-1D could not completely penetrate into the matrix due to the high filament packing density (Figure 21a). Filament bundle disintegration (Figure 21a) and brittle fractured matrix particles on the surface of multiple filaments near the fiber-matrix boundary territory (Figure 21b) were detected. Brittle fractured matrix particles on the filament surfaces (Figure 21c) and multiple inter-filament matrix breakages at the basalt warp (0°) near the boundary area in the B-2D were observed (Figure 21c). In the boundary of the matrix-filament of the B-2D concrete, multiple fractured matrix particles in the filament-matrix at the basalt filling surfaces (90°) were detected (Figure 21d).

Figure 22a–d depicts the SEM graphs of internal fracture surface of the B-4D concrete materials. Sputtered particles of multiple micro matrix breakages on the surface of basalt warps (0° direction) in the B-4D concrete were observed (Figure 22a). Brittle matrix breakages and broom-like damage features were also found. This perhaps indicated that delaying micro-crack propagation during stress transfer phenomena occurred. Filament-matrix debonding and filament pull-out traces on the basalt filling (90°) of the B-4D concrete were realized (Figure 22b). Minor splitting filaments and minor fractured matrix particles on the filament surfaces in the bias (+45°) basalt TOW were observed (Figure 22c) and multiple brittle matrix-filament breakages near the outer region of bias (+45°) basalt TOW, where complete filament-matrix bonding were identified (Figure 22d).

## 4. Implementation of Research Outcome in Real-Life Projects

Multiaxis 3D fiber/matrix concretes can find several applications subjected to aggressive environments. For instance, they can be used as prefabricated panels, bridge deck, industrial floors, seawall construction, substation reactor bases and airport runways [107]. In addition, they can be used for rehabilitation purpose as repairing or retrofitting in ageing infrastructure applications. Some of the examples of such applications are seismic retrofit, confinement, connectors for numerous settlements. This is probably due to their better damage-tolerant material properties based on critical multiaxially load-carrying capacity. This could also be attributed to their structural architecture together with the continuous form of fiber TOW and controls of the multiple micro-crack formation during complex loading.

## 5. Conclusions

Multiaxis 3D basalt fiber/matrix concretes were made. Their flexural properties were empirically investigated and compared to the pristine concrete. Failed interior areal morphologies of basalt fiber concretes were examined by image processing techniques.

The non-interlace preform structures were multiaxially created by placing all continuous filamentary bundles in the in-plane direction of the preform via developed flat winding-molding method.

Four distinct phases of the load-displacement graph in multiaxis basalt concretes were identified: elastic phase, first crack phase, strain-hardening and entire failure phase.

It was discovered that the placement and orientations of the continuous basalt fiber in the pristine concrete exceptionally affected the load-bearing capacity of multiaxis 3D concrete composites (Figure 8 and Figure 9). The flexural strength of the B-1D-(0°), B-2D-(0°, 90° and +45°) and B-4D-(0°, +45° and −45°) concrete composites were considerably greater compared to that of the pristine concrete. Generally, the flexural strength performance on various developed basalt fiber concrete composites was proportional to the fiber orientation and amount of fiber in a particular direction depending upon the structural fiber architecture.

The flexural energy absorption performance of the B-1D-(0°), B-2D-(0° and 90°) and B-4D-(0°, +45° and −45°) concretes were outstandingly higher compared to the control concrete.

Plain concrete exhibited fatal brittle crack. The fracture mechanisms in the four directional basalt concretes (B-4D) were recognized as regional brittle matrix fracture and broom-like damage features on the filaments, filament bundle-matrix debonding for each yarn set and filament bundle splitting along with minor filament entanglement near the core of the B-4D, fiber bridging in the fiber TOWs (maybe particle bridging in the matrix) because of tensile strengthening and immediately after outer filament pull-out and stick-slip between matrix-outer filament or core filament-filament friction were observed. This probably induced multiple micro-cracks in B-4D concretes. However, the numbers of cracks in B-4D and B-2D concretes were greater and multiaxially distributed compared to B-1D concrete composite.

The multiaxis 3D basalt fiber concrete, particularly in the B-4D structure, controlled the number of cracks, crack width and crack distance during four-point flexure loading. Thus, the multiaxis 3D basalt fiber concretes can be regarded as “damage-tolerant materials” compared to the control concrete.

However, additional research studies are required to understand the crack initiation and propagation in the multiaxis 3D concrete volume under complex loading.

## Figures and Tables

**Figure 1 materials-14-02713-f001:**
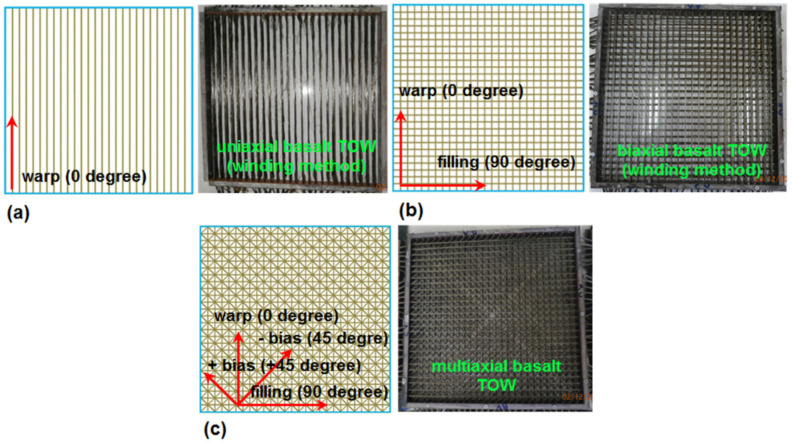
Schematic and actual multiaxis 3D basalt preforms for concrete using the winding-molding method. (**a**) Basalt uniaxial preform (B-1D); (**b**) basalt biaxial preform (B-2D); (**c**) basalt multiaxial preform (B-4D) (digital image).

**Figure 2 materials-14-02713-f002:**
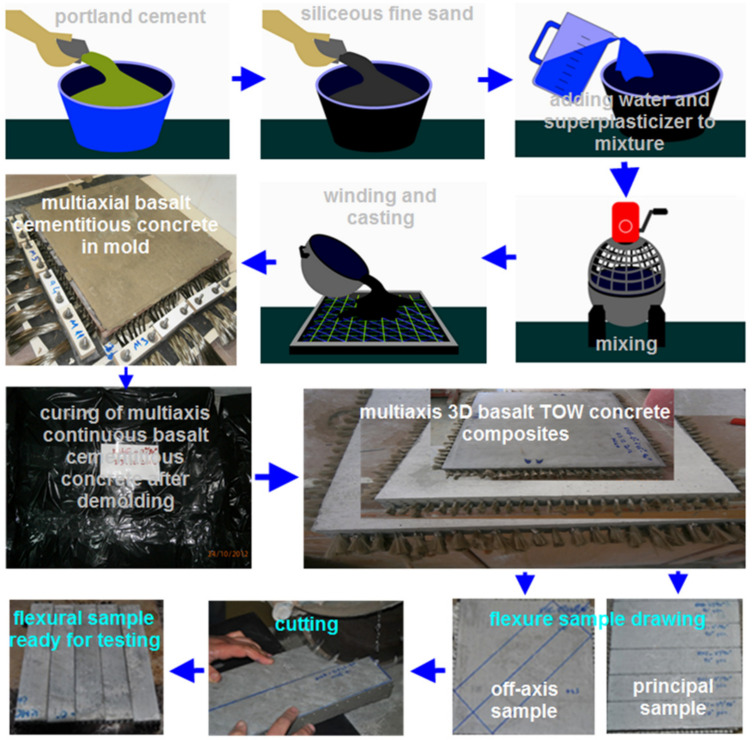
The preparation steps of multiaxis 3D basalt/cementitious concrete, pictorial and schematic views.

**Figure 3 materials-14-02713-f003:**
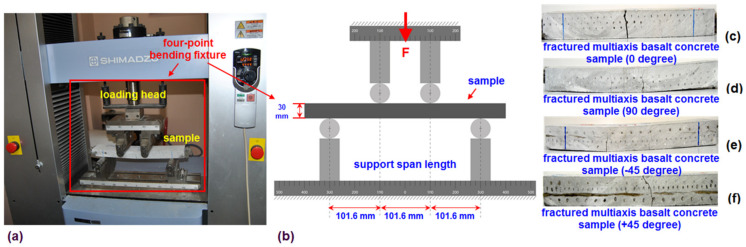
Flexural test of the multiaxis 3D basalt concretes. (**a**) Multiaxis 3D basalt fiber concrete sample in the universal testing instrument; (**b**) specimens with a four-point flexural test fixture, schematic; (**c**) failed basalt concretes, B-4D-(0°); (**d**) B-4D-(90°); (**e**) B-4D-(+45°) and (**f**) B-4D-(−45°), (digital image).

**Figure 4 materials-14-02713-f004:**
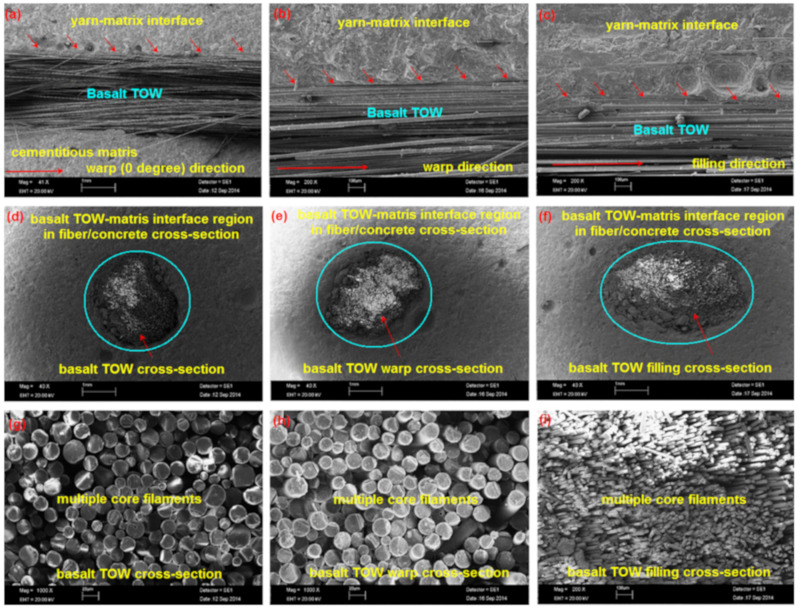
SEM images of some multiaxis 3D basalt concretes before the flexure test. (**a**) B-1D structure in warp length (the scale bar 1 mm, magnification ×41); (**b**) B-2D structure in warp length (100 μm, ×200); (**c**) B-4D structure in filling length (100 μm, ×200); (**d**) B-1D structure in filament bundle cross-section (1 mm, ×40); (**e**) B-2D structure in warp cross-section (1 mm, ×40); (**f**) B-4D structure in filling cross-section (1 mm, ×40); (**g**) basalt filament bundle core of B-1D (20 μm, ×1000); (**h**) close view of the filament bundle core of B-2D (20 μm, ×1000); (**i**) close views of filling filament bundle core of B-4D (100 μm, ×200).

**Figure 5 materials-14-02713-f005:**
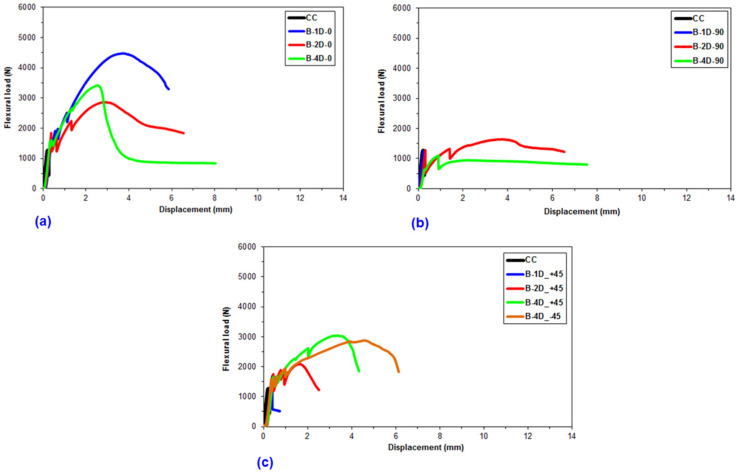
Load-displacement graph of the multiaxis 3D basalt fiber concretes in the four-point flexure. (**a**) Various basalt fiber TOWs concrete structures at 0° flexure; (**b**) various basalt fiber concrete structures at 90° flexure test and (**c**) various basalt fiber concrete structures at bias directions flexure.

**Figure 6 materials-14-02713-f006:**
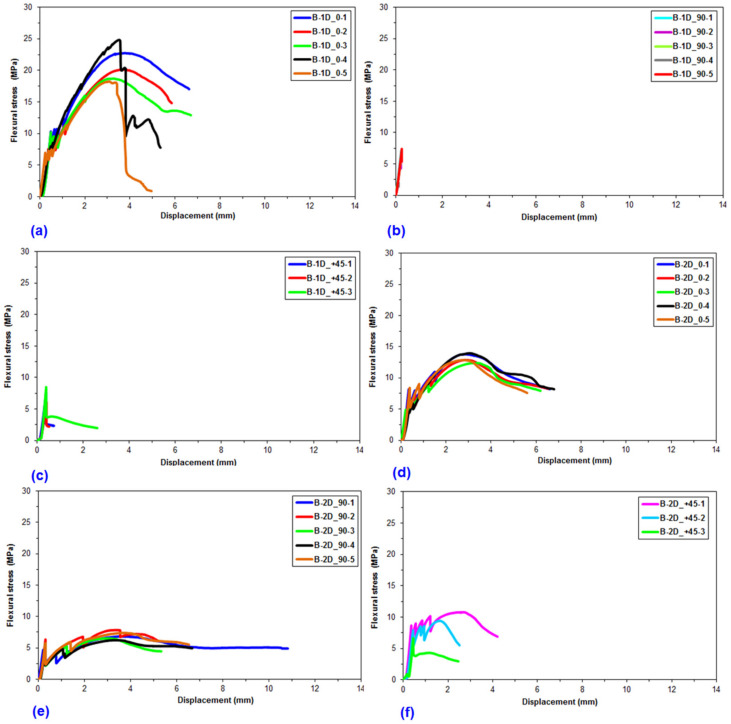
Flexural stress-displacement graph of uniaxial and biaxial 3D basalt concretes. (**a**) B-1D at 0° flexure test; (**b**) B-1D at 90°; (**c**) B-1D at +45°; (**d**) B-2D at 0°; (**e**) B-2D at 90° and (**f**) B-2D at +45°.

**Figure 7 materials-14-02713-f007:**
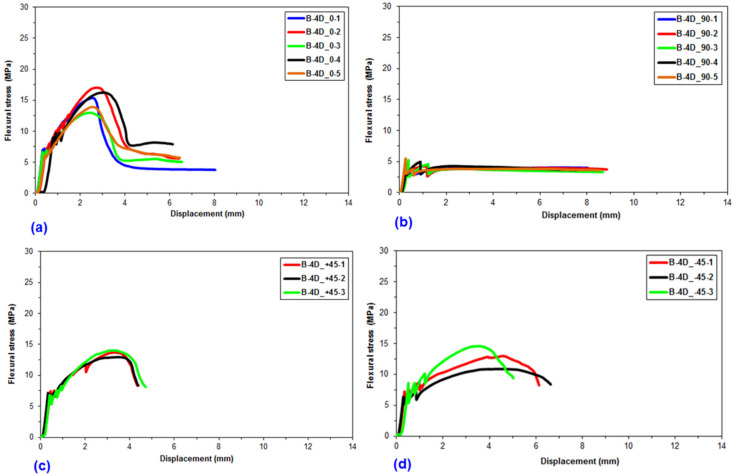
Stress-displacement graph of multiaxis 3D basalt TOW concretes in the four-point flexure test. (**a**) B-4D at 0° flexure test; (**b**) B-4D at 90° flexure test; (**c**) B-4D at +45° direction flexure test; (**d**) B-4D at −45° direction flexure test.

**Figure 8 materials-14-02713-f008:**
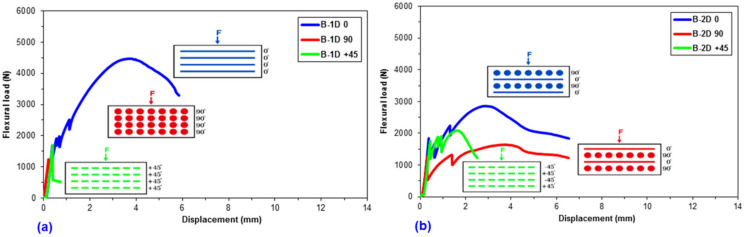
General characterization of the off-axis flexural load-displacement graph of the multiaxis 3D basalt TOWs concretes. (**a**) B-1D concrete composite and (**b**) B-2D concrete composite.

**Figure 9 materials-14-02713-f009:**
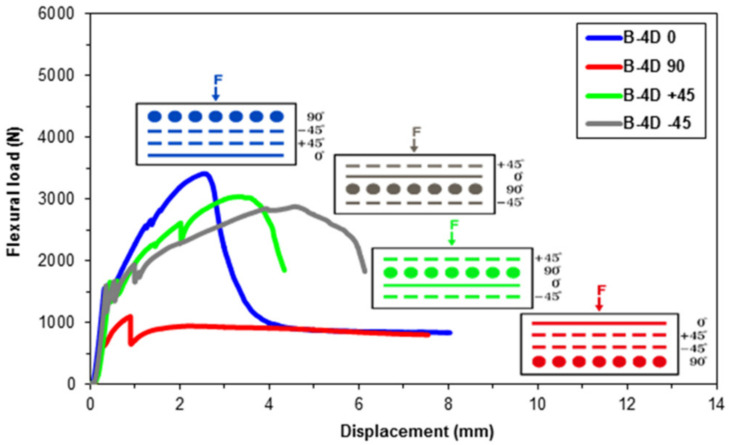
General characterization of the off-axis flexural load-displacement curves of B-4D concrete composite.

**Figure 10 materials-14-02713-f010:**
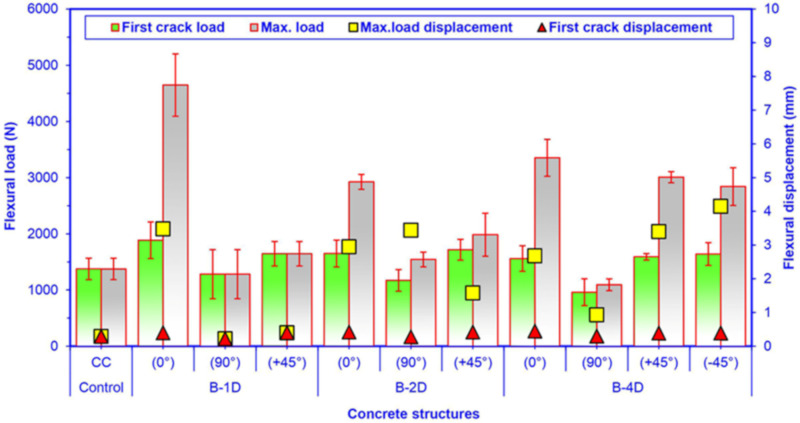
Average maximum off-axis flexure load-displacements and first crack load for all multiaxis 3D basalt fiber concretes.

**Figure 11 materials-14-02713-f011:**
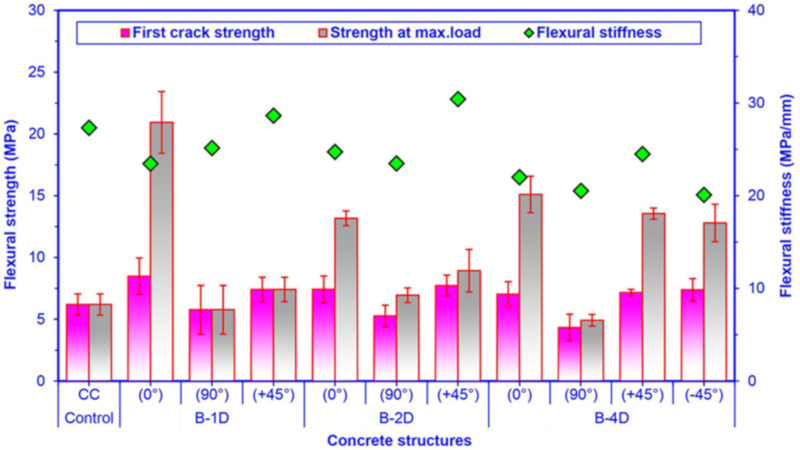
Off-axis flexure properties of all multiaxis 3D basalt concretes.

**Figure 12 materials-14-02713-f012:**
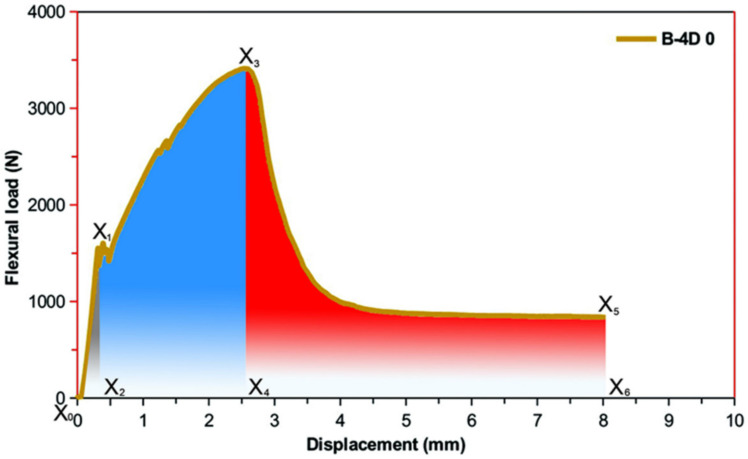
Representative presentation of areas under the load-displacement graph of the multiaxis 3D basalt fiber TOW concretes for the calculation of flexure energy of various stages including first crack, maximum load and strain-hardening.

**Figure 13 materials-14-02713-f013:**
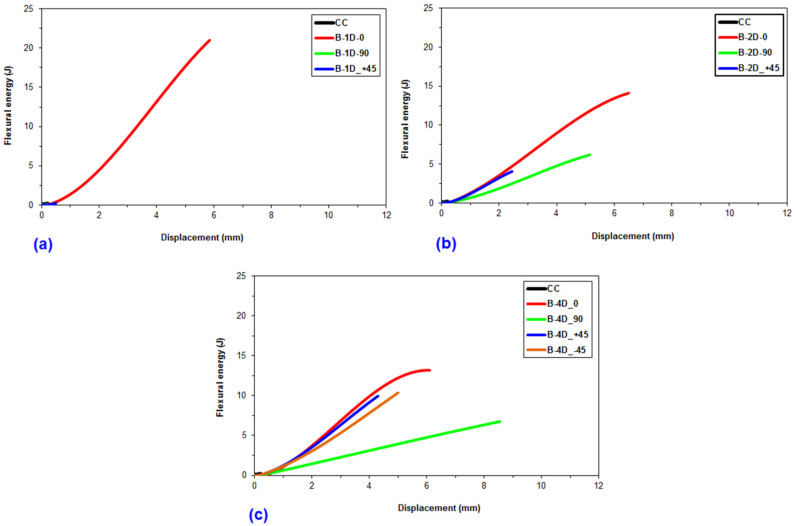
Flexure energy-displacement graphs of the multiaxis 3D basalt fiber TOWs concretes after the off-axis four-point flexure test. (**a**) B-1D concrete; (**b**) B-2D concrete and (**c**) B-4D concrete composites.

**Figure 14 materials-14-02713-f014:**
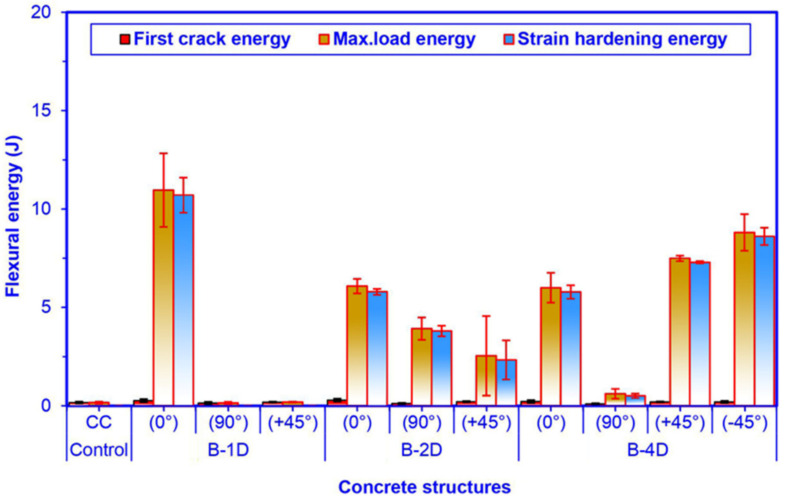
Off-axis flexure energy, first crack energy and strain-hardening energy of all multiaxis 3D basalt fiber TOW concretes.

**Figure 15 materials-14-02713-f015:**
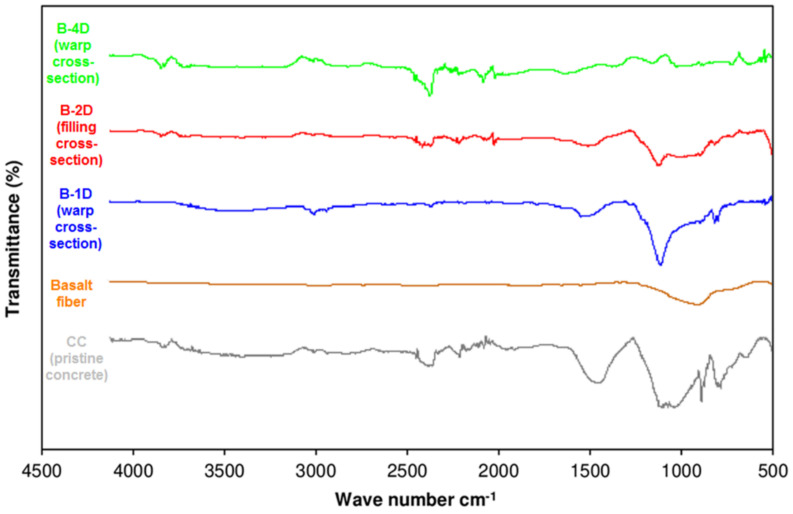
Fourier transform infrared spectra of CC (pristine concrete), basalt fiber, B-1D, B-2D and B-4D concrete composites.

**Figure 16 materials-14-02713-f016:**
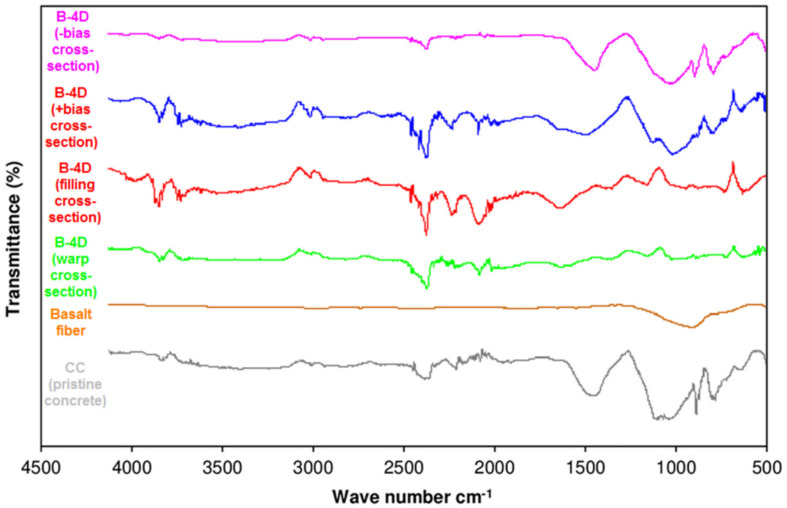
Fourier transform infrared spectra of CC (pristine concrete), basalt fiber and B-4D concrete composites.

**Figure 17 materials-14-02713-f017:**
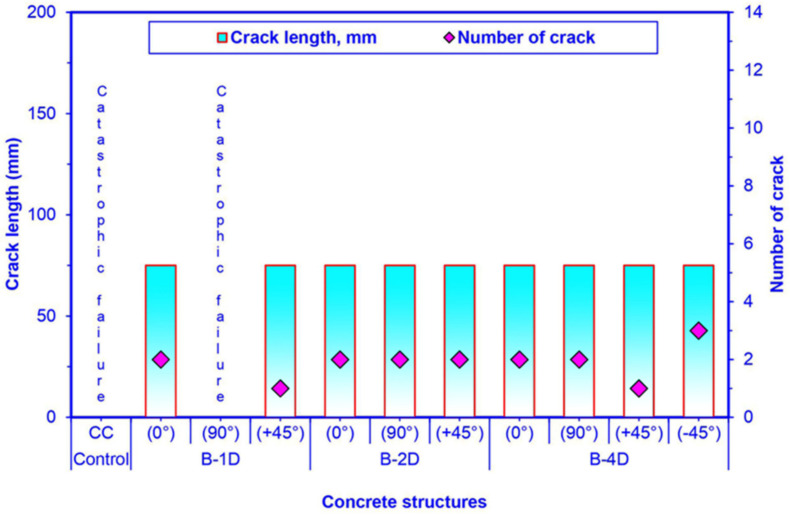
Number of cracks and crack length in fractured multiaxis 3D basalt fiber concretes after off-axis four-point flexure test.

**Figure 18 materials-14-02713-f018:**
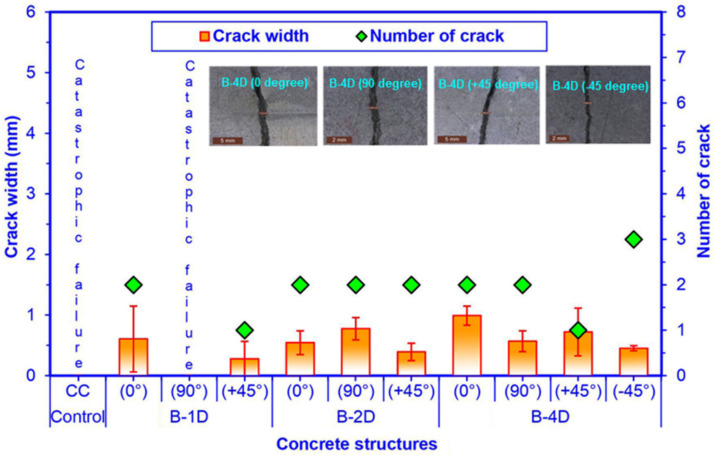
Number of cracks and crack width in fractured multiaxis 3D basalt fiber concretes after off-axis four-point flexure test.

**Figure 19 materials-14-02713-f019:**
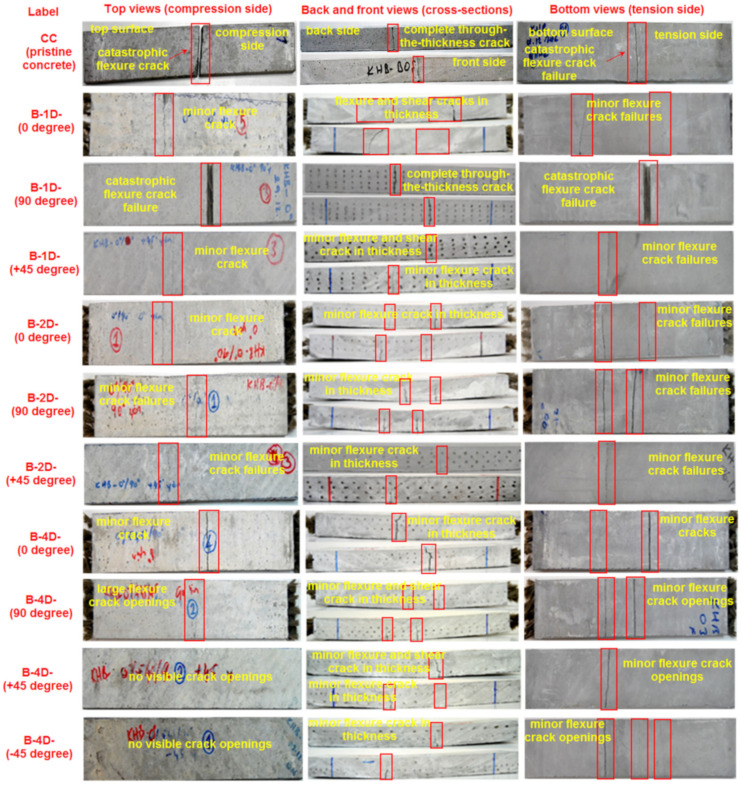
Various fractured multiaxis 3D basalt fibers including neat concretes top, bottom, cross-sections, and their failure modes after the off-axis flexure test (digital photo image).

**Figure 20 materials-14-02713-f020:**
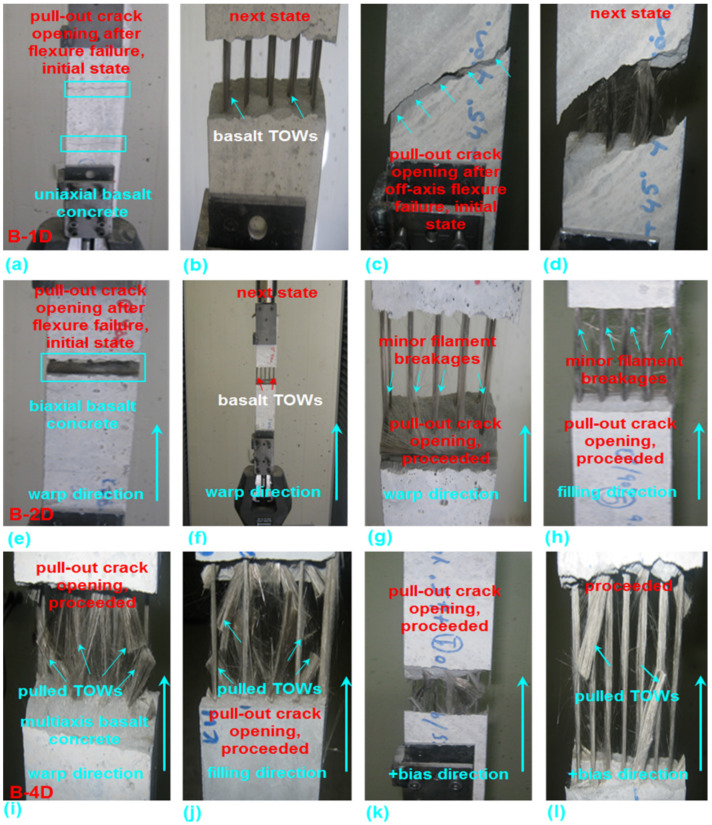
Pull-out of multiaxis 3D basalt concretes. (**a**) Initial state of pull-out crack opening after Figure 1D; (**b**) next state (B-1D); (**c**) initial state (B-1D); (**d**) next state (B-1D); (**e**) initial state (B-2D); (**f**) pull-out fixture (B-2D); (**g**) next step in warp (B-2D); (**h**) next step in filling (B-2D); (**i**) crack opening in warp (B-4D); (**j**) crack opening in filling (B-4D); (**k**) initial state in +bias direction (B-4D) and (**l**) next state in +bias direction (B-4D) (digital photo image).

**Figure 21 materials-14-02713-f021:**
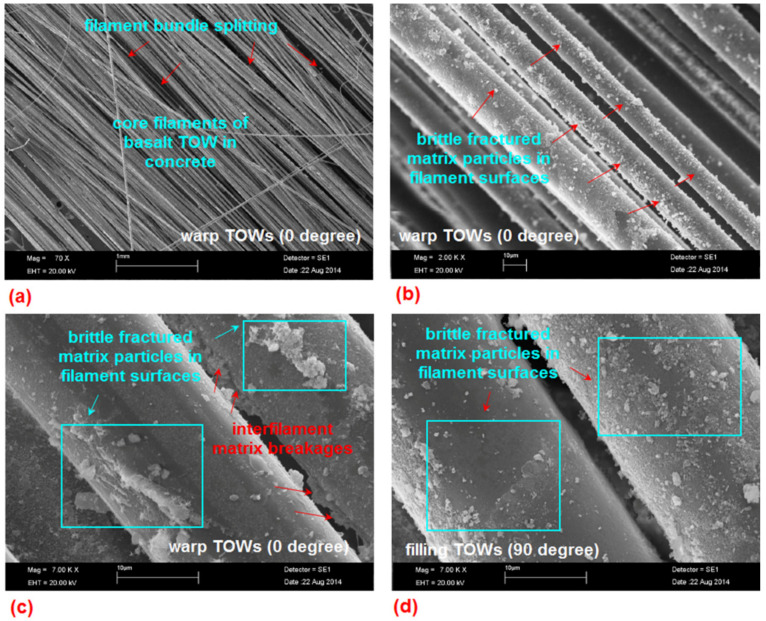
SEM graphs of the failed multiaxis 3D basalt fiber concrete. (**a**) B-1D (the scale bar 1 mm, magnification ×70); (**b**) enlarge view of failed fiber-matrix in B-1D (10 μm, ×2000); (**c**) failed matrix in basalt warp TOW in B-2D (10 μm, ×7000); (**d**) fractured filling fiber TOW in the B-2D concrete (10 μm, ×7000).

**Figure 22 materials-14-02713-f022:**
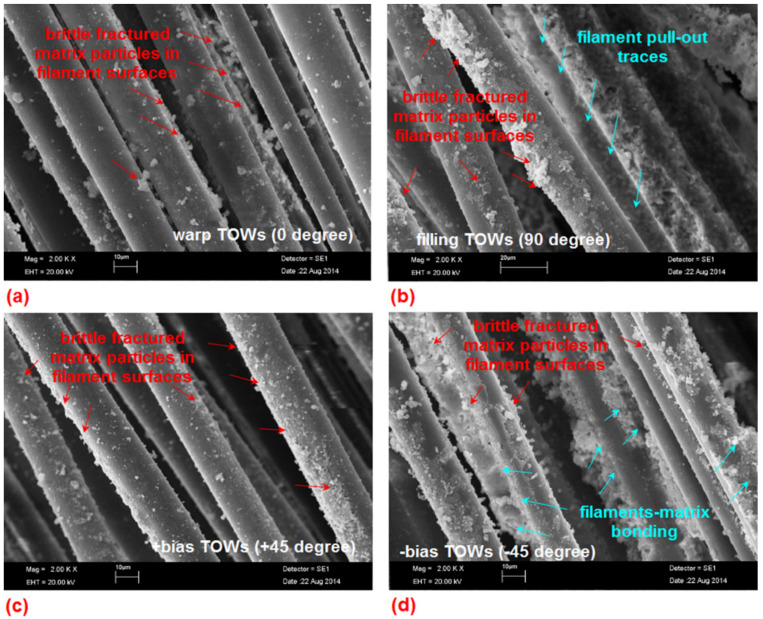
SEM graphs of failed multiaxis 3D basalt fiber concretes. (**a**) Basalt warp surface in the B-4D (the scale bar 10 μm, magnification ×2000); (**b**) enlarged view of the fractured fiber-matrix in filling in B-4D (20 μm, ×2000); (**c**) fractured matrix in bias fiber (+45° direction) in B-4D (10 μm, ×2000); (**d**) fractured matrix in bias fiber (−45° direction) in the B-4D concrete (10 μm, ×2000).

**Table 1 materials-14-02713-t001:** The continuous basalt TOW properties used in basalt fiber concrete composites [87,88].

Fiber	Fiber Diameter (μm)	Linear Density (tex)	Density (g/cm^3^)	Dry Fiber Tenacity (cN/tex)	Tensile Strength (MPa)	Tensile Modulus (GPa)	Breaking Elongation (%)	Melting Temperature (°C)
Basalt (technobasalt, UKR)	16	4800	2.80	50–55 (1400–1540, MPa)	2900	89	3.15	1450

**Table 2 materials-14-02713-t002:** Various multiaxis 3D basalt preforms for the concretes.

Type of Structure	Label	Fiber Type	Yarn Ends	Weave Type	Stacking Sequence/Number of Layers	Number of Twisting (Turn/m)	Plied Yarn (Tex)	Density of Preform (End/15 cm)	Mesh Opening (mm)
Control	CC	Concrete	-	-	-	-	-	-	-
Basalt concrete	B-1D-(0°)	Basalt	4	Non-interlaced	[0°]_4_	10	9600	10	15
	B-1D-(90°)				[90°]_4_				
	B-1D-(+45°)				[+45°]_4_				
	B-2D-(0°)	Basalt	4	Non-interlaced	[90°/0°]_2_	10	9600	10	15 × 15
	B-2D-(90°)				[0°/90°]_2_				
	B-2D-(+45°)				[−45°/+45°]_2_				
	B-4D-(0°)	Basalt	4	Non-interlaced	[90°/±45°/0°]_1_	10	9600	10	15 × 15 ×15 × 15
	B-4D-(90°)				[0°/±45°/90°]_1_				
	B-4D-(+45°)				[+45°/90°/0°/−45]_1_				
	B-4D-(−45°)				[+45°/0°/90°/−45]_1_				

**Table 3 materials-14-02713-t003:** Specifications of off-axis and principle flexural specimen of the multiaxis 3D basalt fiber concretes.

Label	Density (kg/m^3^)	Total Fiber Weight Fractions (%)	Partial Fiber Weight Fraction (%)	Void Content (%)	Water Absorption (%)
CC	2.190	-	-	22.080	11.160
B-1D-(0°)	2.170	3.830	3.830 (0°)	18.045	8.735
B-1D-(90°)	2.170	3.830	3.830 (90°)	18.045	8.735
B-1D-(+45°)	2.170	3.830	3.830 (+45°)	18.045	8.735
B-2D-(0°)	2.210	3.760	1.880 (0°),1.880 (90°)	18.855	9.490
B-2D-(90°)	2.210	3.760	1.880 (90°),1.880 (0°)	18.855	9.490
B-2D-(+45°)	2.210	3.760	1.880 (+45°),1.880 (–45°)	18.855	9.490
B-4D-(0°)	2.220	3.750	0.9375 (90°), 0.9375 (−45°), 0.9375 (+45°), 0.9375 (0°)	19.935	10.060
B-4D-(90°)	2.220	3.750	0.9375 (0°), 0.9375 (+45°), 0.9375 (−45°), 0.9375 (90°)	19.935	10.060
B-4D-(+45°)	2.220	3.750	0.9375 (+45°), 0.9375 (90°), 0.9375 (0°), 0.9375 (−45°)	19.935	10.060
B-4D-(−45°)	2.220	3.750	0.9375 (+45°), 0.9375 (0°), 0.9375 (90°), 0.9375 (−45°)	19.935	10.060

**Table 4 materials-14-02713-t004:** Average flexural results on the multiaxis 3D basalt fiber concretes.

Label	Flexural First Crack Load (N)	Flexural First Crack Displacement (mm)	Flexural Maximum Load (N)	Flexural Maximum Displacement (mm)	Flexural First Crack Strength (MPa)	Flexural Maximum Strength (MPa)	Flexural Stiffness (MPa/mm)
CC	1374.93 ± 190.83	0.29 ± 0.07	1374.93 ± 190.83	0.29 ± 0.07	6.19 ± 0.86	6.19 ± 0.86	27.33 ± 1.08
B-1D-(0°)	1884.49 ± 325.94	0.40 ± 0.09	4646.59 ± 555.50	3.48 ± 0.27	8.49 ± 1.47	20.93 ± 2.50	23.46 ± 4.34
B-1D-(90°)	1281.43 ± 436.76	0.22 ± 0.05	1281.43 ± 436.76	0.22 ± 0.05	5.77 ± 1.97	5.77 ± 1.97	25.15 ± 5.36
B-1D-(+45°)	1645.00 ± 218.25	0.40 ± 0.00	1645.00 ± 218.25	0.40 ± 0.00	7.41 ± 0.99	7.41 ± 0.99	28.65 ± 4.80
B-2D-(0°)	1648.61 ± 239.04	0.42 ± 0.07	2923.85 ± 133.47	2.95 ± 0.19	7.42 ± 1.08	13.17 ± 0.60	24.74 ± 5.20
B-2D-(90°)	1170.53 ± 192.70	0.28 ± 0.04	1542.63 ± 131.39	3.44 ± 0.29	5.27 ± 0.87	6.95 ± 0.59	23.49 ± 1.77
B-2D-(+45°)	1715.52 ± 185.88	0.42 ± 0.04	1983.14 ± 382.57	1.58 ± 0.91	7.73 ± 0.84	8.93 ± 1.72	30.42 ± 2.86
B-4D-(0°)	1559.50 ± 227.30	0.45 ± 0.17	3352.22 ± 328.74	2.68 ± 0.22	7.02 ± 1.02	15.10 ± 1.48	21.99 ± 2.12
B-4D-(90°)	960.61 ± 239.17	0.30 ± 0.05	1092.53 ± 104.86	0.93 ± 0.98	4.33 ± 1.08	4.92 ± 0.47	20.54 ± 3.95
B-4D-(+45°)	1590.08 ± 57.20	0.39 ± 0.05	3007.62 ± 99.04	3.40 ± 0.04	7.16 ± 0.26	13.55 ± 0.45	24.50 ± 2.64
B-4D-(−45°)	1639.72 ± 202.78	0.39 ± 0.09	2840.53 ± 336.03	4.15 ± 0.45	7.38 ± 0.91	12.79 ± 1.52	20.10 ± 3.65

**Table 5 materials-14-02713-t005:** Average flexural energy results in various stages of multiaxis 3D basalt fiber concretes.

Label	Flexural First Crack Energy (J)	Flexural Maximum Load Energy (J)	Flexural Strain-Hardening Energy (J)	Flexural Strain-Softening Energy (J)	Flexural Total Energy (J)
CC	0.16 ± 0.05	0.16 ± 0.05	0.00 ± 0.00	0.01 ± 0.00	0.17 ± 0.05
B-1D-(0°)	0.26 ± 0.08	10.96 ± 1.87	10.70 ± 0.90	8.31 ± 1.44	19.27 ± 4.74
B-1D-(90°)	0.14 ± 0.06	0.14 ± 0.06	0.00 ± 0.00	0.00 ± 0.00	0.14 ± 0.06
B-1D-(+45°)	0.19 ± 0.02	0.19 ± 0.02	0.00 ± 0.00	0.00 ± 0.00	0.75 ± 0.63
B-2D-(0°)	0.29 ± 0.07	6.08 ± 0.37	5.79 ± 0.15	7.88 ± 0.48	13.96 ± 1.32
B-2D-(90°)	0.12 ± 0.03	3.92 ± 0.57	3.80 ± 0.27	4.43 ± 0.89	8.35 ± 2.35
B-2D-(+45°)	0.21 ± 0.03	2.54 ± 2.02	2.33 ± 1.00	2.08 ± 0.30	4.62 ± 2.61
B-4D-(0°)	0.22 ± 0.07	6.00 ± 0.76	5.78 ± 0.35	6.65 ± 0.14	12.65 ± 1.03
B-4D-(90°)	0.10 ± 0.03	0.61 ± 0.25	0.51 ± 0.11	6.23 ± 0.04	6.84 ± 0.32
B-4D-(+45°)	0.20 ± 0.02	7.49 ± 0.14	7.29 ± 0.06	2.88 ± 0.19	10.37 ± 0.51
B-4D-(-45°)	0.20 ± 0.05	8.81 ± 0.93	8.61 ± 0.44	4.49 ± 0.07	13.30 ± 0.80

**Table 6 materials-14-02713-t006:** Failure results on the multiaxis 3D basalt fiber concretes.

Label	Number of Cracks		Crack Length (mm)		Crack Width (mm)		Crack Spacing (mm)
	Top Side	Bottom Side	Topside	Bottom Side	Topside	Bottom Side	Bottom Side
CC	Catastrophic failure	Catastrophic failure	-	75	-	-	-
B-1D-(0°)	-	2	-	75	-	0.604	129.30
B-1D-(90°)	Catastrophic failure	Catastrophic failure	-	75	-	-	-
B-1D-(+45°)	-	1	-	75	-	0.278	0.00
B-2D-(0°)	-	2	-	75	-	0.543	101.30
B-2D-(90°)	-	2	-	75	-	0.774	84.30
B-2D-(+45°)	-	2	-	75	-	0.392	37.70–77.70
B-4D-(0°)	-	2	-	75	-	0.990	109.90
B-4D-(90°)	-	2	-	75	-	0.568	76.80
B-4D-(+45°)	-	1	-	75	-	0.719	0.00
B-4D-(−45°)	-	3	-	75	-	0.451	46.10–53.80

## Data Availability

Data is provided in article.

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
