# Peer review of "Off-Axis Flexural Properties of Multiaxis 3D Basalt Fiber Preform/Cementitious Concretes: Experimental Study"

_materials, 2021, doi:10.3390/ma14112713_

Round 1
Reviewer 1 Report
The authors studied the mechanical properties of multiaxial 3D continuous basalt fibre preform cementitious concrete in this research work. Overall, the paper is well written and meet the technical requirements, however, the quality of the presentation needs to be improved. The authors have done complete literature on the use of basalt fibre in concrete. The novelty of the study needs to be highlighted in the abstract and introduction. In my opinion, this manuscript is acceptable for the Journal of Materials after some minor corrections.
- The abstract should be revised to include the research gap and the novel of the research.
- The abbreviations must be defined before using the text. (what is TOW in the abstract?)
- The authors should call the figures in the text in a consistent way.
- The label of figures is not concise and clear. The authors should use the labels like (a), (b)… under the picture instead of labelling over the images.
- The caption of figures is lengthy (e.g., Figures 3,6 and 20). The explanation can be added to the text.
- There is too much text in some figures (e.g., Figures 19 and 20)
- The authors should be consistent in the size, font and colour of the figures, charts and tables in the whole manuscript. For instance, figure 5, 6, 8 and 13 are very small and difficult to read while in Figures 9 and 12 the font size is too large.
- All units should be checked thoroughly in the text (e.g. degree Celsius in Table 1).
- The scale of SEM images in Figure 4 is not precise.
- The language of the manuscript needs to be polished.
Author Response
RESPONSE TO REVIEWER(S)’ COMMENTS [Manuscript ID: materials-1215366 ]
Date: 13 May 2021
Reviewer comments:
Reviewer 1
Comments and Suggestions for Authors
The authors studied the mechanical properties of multiaxial 3D continuous basalt fibre preform cementitious concrete in this research work.
Answer: This is correct.
Overall, the paper is well written and meet the technical requirements,
Answer: Thank you for this nice comment.
however, the quality of the presentation needs to be improved.
Answer: We can do the required revision to enhance the paper quality.
The authors have done complete literature on the use of basalt fibre in concrete.
Answer: This is almost correct.
The novelty of the study needs to be highlighted in the abstract and introduction.
Answer: Thank you for the comments. The novelty of the study was highlighted in abstract and introduction of the manuscript.
In my opinion, this manuscript is acceptable for the Journal of Materials after some minor corrections.
Answer: Thank you for the comment. We will do the minor corrections in the manuscript.
The abstract should be revised to include the research gap and the novel of the research.
Answer: The research gap and novelty of the research were included in the abstract of the manuscript.
The abbreviations must be defined before using the text. (what is TOW in the abstract?)
Answer: The TOW is not abbreviations in textile science; rather it is definition as “an untwisted bundle of continuous filaments”. It was defined in the first instance related part of the manuscript which is introduction part and highlighted red color.
The authors should call the figures in the text in a consistent way. The label of figures is not concise and clear. The authors should use the labels like (a), (b)… under the picture instead of labelling over the images.
Answer: All Figures in in the text were called consistently and the required revisions on the figure labels were made and highlighted red color.
The caption of figures is lengthy (e.g., Figures 3, 6 and 20). The explanation can be added to the text.
Answer: The revision on the Figure captions 3, 6 and 20 were made and their explanations in caption were shortened.
There is too much text in some figures (e.g., Figures 19 and 20)
Answer: The text in the Figure 19 and 20 were shortened.
The authors should be consistent in the size, font and colour of the figures, charts and tables in the whole manuscript. For instance, figure 5, 6, 8 and 13 are very small and difficult to read while in Figures 9 and 12 the font size is too large.
Answer: The Figures 5, 6, 8 and 13, 9 and 12 as well as Charts and Tables were revised to get consistent in the size, font and color.
All units should be checked thoroughly in the text (e.g. degree Celsius in Table 1).
Answer: All units in Tables were checked carefully especially for units and revised, and highlighted red color.
The scale of SEM images in Figure 4 is not precise.
Answer: The scale of SEM images in Figure 4 is now become precise via providing the amount of scale and magnification in the Figure caption.
The language of the manuscript needs to be polished.
Answer: The required polishing in the manuscript was made and highlighted red color.
We think that the required revisions on the manuscript based on reviewer comments were completed and we would like to do additional revision, if required.
My best regards,

Reviewer 2 Report
The manuscript is potentially interesting to readers of journal Materials.
The problem is current and the manuscript also has potential new knowledge.
Overall, however, the manuscript needs to be reworked and improved in order to increase the informative value of the research.
The manuscript is relatively long. Some information is also known or does not have sufficient informative value.
The research itself is solved logically.
Figure 1. enlarge the picture,
Split Figure 3 into multiple images. The text in the image is very small.
One of the limitations of research is that it only closely specializes in one selected type of test. It would be appropriate to carry out a more extensive experimental program also describing, for example, compressive strength, split tensile strength, modulus of elasticity and application of reinforcement in structural element tests or the use of numerical modeling. From a similar area comprehensively solves area, for example:
Cajka, R .; et. al. Numerical Modeling and Analysis of Concrete Slabs in Interaction with Subsoil. Sustainability 2020, 12, 9868.
It would also be appropriate to expand the introduction section and improve the input to the solved problem. The researched area is given great attention in research.
Figure 4 - complete the scale (zoom) for individual sub-images.
Figures 5, 6, 7, 8 are very small. Curves, descriptions and values cannot be distinguished.
The mathematical notation of equations 4,5,6,7 is not clear. Improve equations and improve description.
Figure 19. - the text in the sub-pictures is very small.
Figure 21, 22 specify scale or zoom.
4. Implementation of research outcome in real-life projects 670
Expand the details and this part
Part of the conclusion must be significantly expanded. The manuscript has a total of 35 pages and new knowledge is summarized on the half page. New knowledge and understanding must be clearly stated in the context of the current situation of research. Part of the discussion, criticism, descriptions of limits of the results must also be expanded.
Overall, the authors spent time working in the experimental program, which, however, requires informative value for readers and further research.
The manuscript must be revised.
Author Response
RESPONSE TO REVIEWER(S)’ COMMENTS [Manuscript ID: materials-1215366 ]
Date: 13 May 2021
Reviewer comments:
Reviewer 2
Comments and Suggestions for Authors
The manuscript is potentially interesting to readers of journal Materials.
Answer: Thank you for this nice comment.
The problem is current and the manuscript also has potential new knowledge.
Answer: Thank you for the comment. I agree with reviewer 2.
Overall, however, the manuscript needs to be reworked and improved in order to increase the informative value of the research.
Answer: The required revision can be made to increase the scientific value of the manuscript.
The manuscript is relatively long. Some information is also known or does not have sufficient informative value. The research itself is solved logically.
Answer: We would like to keep all knowledge provided in the manuscript due to sequential explanation for readers stand points, if possible.
Figure 1. enlarge the picture,
Answer: The Figure 1 was enlarged.
Split Figure 3 into multiple images. The text in the image is very small.
Answer: The text in Figure 3 was magnified and became large to read them easily.
One of the limitations of research is that it only closely specializes in one selected type of test.
Answer: This is partly true that this research output concentrated on the off-axis flexure properties. However, we originally investigated multiaxis concrete composite’s panel properties, flexural properties and most importantly pull-out properties which identify the interphase behavior of fiber and cement. We will publish those properties separately due to paper becomes very long.
It would be appropriate to carry out a more extensive experimental program also describing, for example, compressive strength, split tensile strength, modulus of elasticity and application of reinforcement in structural element tests or the use of numerical modeling.
Answer: This is true that more properties should be investigated as reviewer 2 mentioned the compressive strength, split tensile strength and tensile modulus and applications of the structures in real life as well as numerical modeling. We developed the multiaxis 3D fiber/cementitious concrete composite and their flexure and panel and pull-out properties were studied. However, for future endeavor, we will investigate those properties including numerical modeling. Right now, we have no research money, no student and no critical infrastructure facilities.
From a similar area comprehensively solves area, for example: Cajka, R .; et. al. Numerical Modeling and Analysis of Concrete Slabs in Interaction with Subsoil. Sustainability 2020, 12, 9868. It would also be appropriate to expand the introduction section and improve the input to the solved problem. The researched area is given great attention in research.
Answer: The reference suggested by reviewer 2 was analyzed and it included in the introduction of the manuscript and highlighted res color.
Figure 4 - complete the scale (zoom) for individual sub-images.
Answer: The scale in Figure 4 was provided in the Figure caption and highlighted red color.
Figures 5, 6, 7, 8 are very small. Curves, descriptions and values cannot be distinguished.
Answer: Figures 5, 6, 7 and 8 were enlarged and were rearranged to distinguish the curves properly.
The mathematical notation of equations 4,5,6,7 is not clear. Improve equations and improve description.
Answer: The mathematical notation of equations 4, 5, 6 and 7 were revised and becomes clear.
Figure 19. - the text in the sub-pictures is very small.
Answer: The texts in the sub-pictures of Figure 19 were modified and become comparatively large.
Figure 21, 22 specify scale or zoom.
Answer: The scale bar of Figures 21 and 22 were specified in the Figure captions.
- Implementation of research outcome in real-life projects 670
Expand the details and this part
Answer: The expansion of the implementation of research outcome in real-life projects were made and highlighted red color.
Part of the conclusion must be significantly expanded. The manuscript has a total of 35 pages and new knowledge is summarized on the half page. New knowledge and understanding must be clearly stated in the context of the current situation of research.
Answer: The generated new knowledge provided to expand the conclusion significantly.
Part of the discussion, criticism, descriptions of limits of the results must also be expanded.
Answer: The shortcoming of the developed structures on the flexure results were also discussed in the conclusion of the manuscript
Overall, the authors spent time working in the experimental program, which, however, requires informative value for readers and further research.
Answer: I think that the finding from this study on the developed new concrete structure which is called “Textile Reinforced Concrete” is mostly quite informative for readers stand points. However, more study is required bring these concrete structures in the construction practice. For this reasons, future work study was also provided in the conclusion of the manuscript for engineer and researcher who study on the similar subject.
The manuscript must be revised.
Answer: The required revisions suggested by reviewer 2 were carefully made throughout the manuscript.
We think that the required revisions on the manuscript based on reviewer comments were completed and we would like to do additional revision, if required.
My best regards,

Reviewer 3 Report
The authors present a work on the Off-axis Flexural Properties of Multiaxis 3D Basalt Fiber Preform/Cementitious Concretes: Experimental Study. The subject of the authors work is an important significant issue in structural engineering and materials.
The article begins with a literature review that is written correctly in terms of content. It contains interesting information relevant to the topic of the article. The literature list is impressive and contains as many as 116 items. On the plus side, a very large part of the literature is very recent items from the last few years.
Next, the authors present the scope of the study and the methodology used. Here also everything is done very correctly. The preparation of the samples and the conduct of the tests are described very precisely and understandable for the reader.
The analysis of the results is done correctly. I have no comments to it. All figures and diagrams in the article are clear and well prepared. Only remark I have to figure 19. I think that photo of samples shown in this figure are not very clear. Some of them do not show details marked by the authors with a blue frame.
The conclusions of the study are also formulated correctly.Also, the discussion presented is done factually correct and perfectly shows the results and conclusions of the research drawn by the authors. I have no substantive comments to them. The article is very important from the point of view of construction practice. However, I have a question for the authors. Have the results of research conducted by authors been implemented into construction practice? If they were implemented, please briefly describe where and what were the effects? The authors in section 4 only gave information about the possibility of applying the obtained research in practice.
In conclusion, I believe that the article is generally written correctly and does not require any changes. In my opinion, the work presented to me for review exceeds the standards for articles because it contains as many as 35 pages. It could as well be a part of some book or scientific monograph. I think that without any significant corrections it can be published in an international journal such as Materials.
Author Response
RESPONSE TO REVIEWER(S)’ COMMENTS [Manuscript ID: materials-1215366 ]
Date: 13 May 2021
Reviewer comments:
Reviewer 3
Comments and Suggestions for Authors
The authors present a work on the Off-axis Flexural Properties of Multiaxis 3D Basalt Fiber Preform/Cementitious Concretes: Experimental Study. The subject of the authors work is an important significant issue in structural engineering and materials.
Answer: These are correct and thank you for the comment.
The article begins with a literature review that is written correctly in terms of content. It contains interesting information relevant to the topic of the article. The literature list is impressive and contains as many as 116 items. On the plus side, a very large part of the literature is very recent items from the last few years.
Answer: Thank you for reviewer 3. These comments were all correct and precise.
Next, the authors present the scope of the study and the methodology used. Here also everything is done very correctly. The preparation of the samples and the conduct of the tests are described very precisely and understandable for the reader.
Answer: Thank you for these nice and correct comments.
The analysis of the results is done correctly. I have no comments to it. All figures and diagrams in the article are clear and well prepared.
Answer: Thank you for the comment.
Only remark I have to figure 19. I think that photo of samples shown in this figure are not very clear. Some of them do not show details marked by the authors with a blue frame.
Answer: The reason is that we showed many textile reinforced concretes all together in one frame. But, the individual figures were getting smaller in order to fit the page. Splitting the figures as small portion prevent us to see the exact failure differences on each developed structures. However, we did some minor revision on the Figure 19 to improve the quality of the images for reader stand points.
The conclusions of the study are also formulated correctly. Also, the discussion presented is done factually correct and perfectly shows the results and conclusions of the research drawn by the authors.
Answer: Thank you very much for the reviewer 3.
I have no substantive comments to them. The article is very important from the point of view of construction practice. However, I have a question for the authors. Have the results of research conducted by authors been implemented into construction practice?
Answer: No. However, the background of this project is that the multiaxis 3D fiber/cementitious concrete composite structure were studied by me (corresponding author) for repairing of aging infrastructures of USA (bridging, airport, nuclear plant, dam etc.) when I worked one of the startup companies in USA. However, the startup company was bankrupt due to bad management and I left the company. Since then, I keep on working the project myself as a personal curiosity. When I got PhD students, the project was transformed from idea to concept and materialized as a form of concrete composite. Probably, structure was promising for several areas related to civil engineering especially seismic loading because of the attractive fracture toughness properties due to continuous filamentary TOW reinforcement in the concrete which can be considered as “damage tolerance material”.
If they were implemented, please briefly describe where and what were the effects?
Answer: The structure is currently under evaluation stage. One of the companies is from Germany asking for earthquake application. But, we have to generate data under the seismic loading. Research in the study will be concentrated on multi-functionalized the structure to get diverse applications as thermal, electromagnetic interference, insulation effectiveness for future.
The authors in section 4 only gave information about the possibility of applying the obtained research in practice.
Answer: This is true that possible application areas for multiaxis 3D fiber/cementitious concrete composite were mentioned in the section 4 of the manuscript.
In conclusion, I believe that the article is generally written correctly and does not require any changes.
Answer: Thank you for the decision.
In my opinion, the work presented to me for review exceeds the standards for articles because it contains as many as 35 pages. It could as well be a part of some book or scientific monograph. I think that without any significant corrections it can be published in an international journal such as Materials.
Answer: Thank you for the decision and thank you for these nice comments.
We think that the required revisions on the manuscript based on reviewer comments were completed and we would like to do additional revision, if required.
My best regards,

Round 2
Reviewer 2 Report
Thank you for the adjustments made.
The changes made the improvement of the manuscript.
There are currently 11 self-citations in the manuscript. Use only very important in the context of the manuscript (max. 5).
The research area and results are from the context of the manuscript can better understand.
The manuscript contains all the main information.
The manuscript can be published in the journal.
Author Response
RESPONSE TO REVIEWER(S)’ COMMENTS [Manuscript ID: materials-1215366 ]
Date: 15 May 2021
Reviewer comments:
Reviewer 2
Comments and Suggestions for Authors
Thank you for the adjustments made.
Answer: You are welcome and thank you for the comments.
The changes made the improvement of the manuscript.
Answer: Thank you for the comments.
There are currently 11 self-citations in the manuscript. Use only very important in the context of the manuscript (max. 5).
Answer: The self-citations were reduced from 11 to 5 and changes in the manuscript were highlighted red color.
The research area and results are from the context of the manuscript can better understand.
Answer: Thank you for this nice comment.
The manuscript contains all the main information.
Answer: Thank you for the comment.
The manuscript can be published in the journal.
Answer: Thank you for the decisions.
My best regards,
